# Radioprotective and Antimutagenic Effects of *Pycnanthus angolensis* Warb Seed Extract against Damage Induced by X rays

**DOI:** 10.3390/jcm9010006

**Published:** 2019-12-18

**Authors:** Daniel Gyingiri Achel, Miguel Alcaraz-Saura, Julián Castillo, Amparo Olivares, Miguel Alcaraz

**Affiliations:** 1Applied Radiation Biology Centre, Radiological and Medical Sciences Research Institute, Ghana Atomic Energy Commission, Legon, Accra GE-257-046, Ghana; gachel@gmail.com; 2Radiology and Physical Medicine Department, School of Medicine, University of Murcia, 30100 Espinardo, Murcia, Spainamparo.o.r@um.es (A.O.); 3Nutrafur S. A., Camino Viejo de Pliego, Km.2, 30820 Alcantarilla, Murcia, Spain; j.castillo@Nutrafur.com

**Keywords:** micronuclei, radioprotectors, radiation effects, melanoma, PNT2, B16F10 cells

## Abstract

Although different studies have demonstrated different applications of *Pycnanthus angolensis* extracts in traditional African and Asian medicine, its possible antimutagenic or genoprotective capacities have never been explored. We studied these capabilities of *Pycnanthus angolensis* seed extract (PASE) by means of the two micronucleus assays, determining the frequency of micronucleus (MN) yield in mouse bone marrow (in vivo) and in human lymphocytes blocked by cytochalasin B (in vitro). PASE exhibited a significant genoprotective capacity (*p* < 0.001) against X-rays with a protection factor of 35% in both in vivo and in vitro assays. Further, its radioprotective effects were determined by the 3-(4,5-dimethyl-2-thiazolyl)-2,5-diphenyl-tetrazolium bromide (MTT) cell viability test in two cell lines: one being radiosensitive (i.e., human prostate epithelium (PNT2) cells) and the other being radioresistant (i.e., B16F10 melanoma cells). In the radiosensitive cells, PASE showed a protection factor of 35.5%, thus eliminating 43.8% of X-ray-induced cell death (*p* < 0.001) and a dose reduction factor of 2.5. In the radioresistant cells, a protection factor of 29% (*p* < 0.001) with a dose reduction factor of 4 was realized. PASE elicited a greater radioprotective capacity than the substances currently used in radiation oncology and, thus, could be developed as a nutraceutical radioprotectant for workers and patients exposed to ionizing radiation.

## 1. Introduction

Numerous studies have portrayed the varied medical applications of different extracts of *Pycnanthus angolensis* in traditional African and Asian medicine [1,2,3,4,5,6]. Extracts of *P. angolensis* have been used as antibacterial [1,3] antiparasitic [4], anti-inflammatory and analgesic [1,3,4,5], and as antihemorrhagic agents [4]. There are also reports about its use as an antidote against poisons [4], for hyperglycemia [1,2], and even against female sterility [1]. References on its antimutagenic/antigenotoxic or radioprotective capacity is rare even though some authors suggest that its antioxidant potential could explain some of the applications described [2].

Our attention was drawn to *P. angolensis* because of the potential applications that may be derived from its suggested potent antioxidant and free radical scavenging capacities [2]. The ability of antioxidants to eliminate reactive oxygen species (ROS) produced by oxidative stress during exposure to ionizing radiation (IR) is considered a protection mechanism against cellular damage induced by IR both in vitro and in vivo [7,8,9,10,11,12].

Thus, in this study, we examined the radioprotective and antimutagenic potentials of *Pycnanthus angolensis* seed extract (PASE) both in vitro and in vivo and compared it with other compounds with known radioprotective properties. This aimed at assessing if the extract offers some level of protection to normal tissues and may uncover novel substances with protection for workers occupationally exposed to ionizing radiation and/or for patients undergoing diagnostic radiology.

## 2. Materials and Methods

### 2.1. Plant Material

Seeds of *P. angolensis* Warb were harvested from plants growing in the wild in cocoa farms in the eastern region of Ghana and authenticated by an established curator at the Ghana Herbarium, Department of Botany, University of Ghana, Accra. The seeds were picked during the maturity period in November 2015 and were initially dried under shade and finally under vacuum at ambient temperature to a moisture content of less than 5% measured as described in the European Pharmacopoeia version 7.0.

### 2.2. Seed Extraction

Dried seeds were chopped into small pieces, mixed with SiO_2_ and diatomaceous earth (10:1:9 respectively), and ground in a laboratory mill to an average particle size of 1 µm. Two hundred grams of this ground material (equivalent to 100 g of seed) was extracted in 99% methanol at a ratio of 10% (w/v) at room temperature (range 22–24 °C) on a Heidolph mechanical stirrer (RZR 2020, Heidolph Instruments, Schwabach, Germany) for 1 h.

The extract was filtered by vacuum filtration through a Büchner funnel attached to a Kitasato flask fitted with a polypropylene filter cloth. The marc was squeezed to recover all the solvent, and the funnel was washed with fresh methanol (200 mL). The volume of extract recovered (1.924 mL) was concentrated under reduced pressure at 40–50 °C in a Heidolph rotary evaporator (Laborota 4000, Heidolph Instruments, Schwabach, Germany) to approximately 1/12th of its original volume, yielding 150–160 mL of a turbid, semiviscous, syrupy liquid. This syrup was mixed with 30 mL (5:1 ratio) of deionized water, prompting flocculation, that was clarified by filtration using a system comprising a Büchner-Kitasato filter system fitted with a one micron cellulose/silica filter membrane (filter plate AF100, Ref 2036-Filtrox, St. Gallen, Switzerland). The clarified extract was concentrated under vacuum at 40–45 °C using a Heidolph rotary evaporator (Laborota 4000) to yield 36 mL of a very dark brown viscous syrup.

This material was extracted in ether 1:7 (*v*/*v*) at room temperature (22–24 °C) for 3 h in a 500 mL Erlenmeyer flask with continual stirring on a Heidolph mechanical stirrer (model RZR 2020, Heidolph Instruments, Schwabach, Germany). The extraction process was repeated thrice, ensuring that the organic phase showed no taint of yellow-orange color. The resulting solutions were pooled and allowed for liquid–liquid partitioning in a 500 mL separatory funnel for one hour. This yielded a 737 mL ethereal layer that was evaporated to dryness at 30–35 °C on a Heidolph rotary evaporator, (Laborota 4000) affording 16.34 g of a dark brownish, oily, and viscous semisolid as final product (PASE).

### 2.3. Chemicals and Reagents

*P. angolensis* seed extract (PASE) was extracted from the seeds as described above. RPMI 1640, Ham’s F10, phytohemagglutinin A (PHA), cytochalasin B, streptomycin, penicillin, phosphate-buffered saline (PBS), 3-(4,5-dimethyl-2-thiazolyl)-2,5-diphenyl-tetrazolium bromide (MTT), vitamin E (δ-tocopherol) (T), bovine serum albumin (BSA fraction V), and fetal bovine serum (FBS) were obtained from Gibco (USA). Glacial acetic acid and ethanol were obtained from Scharlao SL (Madrid, Spain). Methanol and methanol HPLC grade were obtained from Panreac (Madrid, Spain); 5% sodium heparin was obtained from Rovi Pharmaceutical Laboratories (Madrid, Spain).

Rosmarinic acid (RA), diosmin (D), and quercetin (Q) were obtained from Extrasynthese S.A. (Genay, France). Eriodictyol (E) and ascorbic acid (C) were obtained from Sigma-Aldrich Chemicals SA (Madrid, Spain). Dimethyl sulfoxide (DMSO) was obtained from Merck (Darmstadt, Germany). Amifostine (AMF) (Ethyol^®^) was obtained from Schering-Plough S.A (Madrid, Spain). Green tea extract (Te), carnosic acid (CA), and apigenin (API), were supplied by Nutrafur S.A. (Alcantarilla, Murcia, Spain).

### 2.4. Preparation of Plant and Plant Seed Extract (PASE) for Chromatographic Analysis

Active compounds from different seeds of the plant (PASE) were extracted for analytical chromatography using HPLC-grade methanol in the ratio of 20 and 4 mg/mL, respectively. The extraction was done at 25 °C during 30 min in a stirred flask. All solutions were filtered through a 0.45 µm nylon filter membrane before undertaking HPLC analysis. The samples were aliquoted into small vials and stored at 4 °C until required.

#### 2.4.1. HPLC Analysis of *P. angolensis* Seeds and *P. angolensis* Seed Extract (PASE)

Analyses were performed on an HP 1100 liquid chromatographic system (Hewlett- Packard, Waldbronn, Germany) series equipped with an LC-6A double pump (Shimadzu Corporation, Kyoto, Japan). The chromatograms were monitored by a UV–vis diode array detector at a wavelength of 250 nm. The stationary phase was a 250 × 4 mm id., 5 µm, C_18_ reversed-phase HPLC column (Shimadzu Shim-Pack CLC (M)) thermostated at 30 °C. The flowrate was 1 mL/min.

Analysis was made using a gradient between mobile phase A (1% acetic acid) and phase B (methanol) at a flow rate of 1 mL/min as follows: 0–5 min, 50% of A and B; 5–25 min, 50% A and B; 25–35 min, 100% B. The column was finally re-equilibrated with the initial solvent for 5 min (total time 40 min). The main compounds were identified by comparison of their retention times and UV spectra obtained with the diode-array detector with standard compounds. Three experiments were conducted on each sample, while solvent blanks were intermittently injected into columns (column washing) to eliminate peak splitting or tailing during the analysis.

#### 2.4.2. Identification of Extracted Compounds

Further analysis was performed on an Agilent 1100 series HPLC (Agilent Technologies, Germany) coupled to an ion trap VL mass spectrometer detector (Agilent Technologies, Germany) to confirm the identities of the compounds present in the raw seed material and the PASE extract.

The separation was executed on a Waters SunFire^®^ C_18_ column (5 μm particle size and column dimensions of 120 µm, 150 mm × 4.6 mm i.d.). The mobile phase was an isocratic system composed of 20:80 water/methanol and 1% acetic acid, at a flow rate of 0.8 mL/min, thermostated at 30 °C and a total run time of 20 min.

Two signals were acquired, one with a diode array detector at a wavelength of 280 nm and one in the range of 190–380 nm. The mass spectrometer was operated in a scan mode with the electrospray (ESI) source in the positive ion mode (ESI +), a mass detection range of 100–800 amu, and a target mass of 400 m/z. The optimized conditions were nebulizer pressure of 60 psi, a drying gas flow rate of 9 mL/min, and a gas temperature of 350 °C.

#### 2.4.3. Chromatographic Analysis of Used Flavonoids and Polyphenols

HPLC analyses were conducted to confirm the concentrations of active compounds in all flavonoids, polyphenols, and plant extracts used in this structural comparative study (D, Q, Te, and API served as flavonoids; CA and RA as diterpenic caffeoyl compounds respectively). All compounds were dissolved in DMSO at a concentration of 1 mg/mL. All solutions were filtered through a 0.45 µm nylon filter membrane before undertaking HPLC analysis.

Analyses were performed in an HP 1100 liquid chromatographic system (Hewlett- Packard, Waldbronn, Germany) series equipped with an LC-6A double pump (Shimadzu Corporation, Kyoto, Japan). The chromatograms were monitored simultaneously by a UV–vis diode array detector at 280 and 340 nm. The stationary phase was a 250 × 4 mm id., 5 µm, C_18_ reversed-phase HPLC column (Shimadzu Shim-Pack CLC (M)) thermostated at 30 °C. The flow-rate was 1 mL/min.

The following mobile phases were used for chromatographic analysis: (A) acetic acid/water (2.5:97.5) and (B) acetonitrile. A linear gradient was run from 95% (A) and 5% (B) to 75% (A) and 25% (B) during 20 min; changed to 50% (A) and (B) for 20 min (40 min total run time); and changed to 20% (A) and 80% (B) for 10 min (50 min total run time). The column was re-equilibrated for 10 min in the initial solvent (60 min run time).

### 2.5. Genoprotective Studies

#### 2.5.1. Micronucleus Test (CBMN)

Venous blood was obtained by venipuncture from the arm veins of three supposedly healthy young female donors into heparinized tubes. Twenty microliters (20 µL) of a 20 µM concentration of the test substances was added to 2 mL of the heparinized human blood samples at two different times: immediately before exposure to X-rays (treatment before irradiation) or immediately after exposure to the X-rays (post irradiation treatment). Immediately after irradiation with X-rays, the cytokinesis-block micronucleus (CBMN) assay, as described by Fenech and Morley [13] and adapted by the International Atomic Energy Agency [14], was used to access damage in the cultured irradiated human lymphocytes. The number of micronuclei in at least 3000 CB cells for each treatment was determined by three specialists who analyzed the slides using optical microscopes in a double-blind study.

#### 2.5.2. Micronucleus Assay in Mouse Bone Marrow (PCEs)

In the in vivo experiments, male Swiss mice 12 weeks old, distributed in groups of 6 for each of the substances tested, with weights ranging from 27 to 35 g were used. All solutions were prepared daily, and the test substances were dissolved to a concentration of 0.2% in their drinking water. This treatment commenced one week prior to X-ray exposure.

The animals were housed in the Animal Service Laboratory of the University of Murcia (REGAES300305440012), and the procedures used were approved by the Ethical Committee of the Autonomous Community of the Region of Murcia (Spain) (CECA:510/2018).

An in vivo micronucleus assay was performed on the bone marrow of the mice, as described by Schmid [15]. Twenty-four hours after X-ray exposure, the numbers of micronucleated polychromatic erythrocytes (MNPCEs) among 1000 PCEs per mouse were determined by three specialists in a double-blind study. To ensure that the substances tested were nontoxic, the number of normochromatic erythrocytes and of total erythrocytes in each animal were also determined.

### 2.6. Radioprotective Effects: Cell Lines and Culture Conditions

In this study, two cell types selected based on their radiosensitivity status were used: cells traditionally considered radiosensitive (PNT2) and B16F10 cells, which are traditionally considered to be very radioresistant [16]. The normal epithelium prostatic cell line (PNT 2) was obtained from the European Collection of Cell Cultures (ECACC, Salisbury, UK), Health Protection Agency, Culture Collection (catalogue no.:95012613, Salisbury, UK). The PNT2 cells were cultivated in RPMI 1640 (Sigma-Aldrich, Madrid, Spain) supplemented with 10% FBS, 2 mM glutamine, and streptomycin and penicillin (100 μg/mL and 100 IU/mL respectively). The mouse metastatic melanoma cell (B16F10) line was kindly provided by Dr. V. Hearning (NIH, Bethesda, MA, USA) and cultured in Dulbecco’s modified Eagle´s medium (DMEM)/F12K (1:1) (Sigma-Aldrich, St. Louis, MI, USA) supplemented with 10% FBS (Gibco, BRL, Louisville, KY, USA), 4 mM l-glutamine, penicillin (100 IU/mL), and streptomycin (100 µg/mL).The cultures were maintained at 37 °C, a relative humidity of 90%–95%, and an atmosphere of 7.5% CO_2_. Tests were carried out to confirm the absence of *Mycoplasma* spp. throughout the study.

#### Radioprotective Effects: (MTT) Test

To analyze the radioprotective effects of the substances on PNT2 and B16F10 cell lines, two MTT assay types, as previously described [17,18], were carried out. One of these tests lasting 24 h was used to assess cytotoxicity, and the other lasting 48 h was to evaluate cell proliferation after treatment with test substances with and without exposure to X-ray. Briefly, the cultures were incubated in 200 µL growth medium and allowed to adhere for 24 h after cell seeding in both types of assays so cells could adapt to the culture conditions and adhere to the bottom of the wells. For the PNT2 cells 3200 cells/wells and for B10F16 2500 cells/well were established as optimal cell seeding concentrations. Different concentrations of the test substances to be assayed were put into each well (at least 6 wells per test substance), and the plates were exposed to different doses of X-rays (0, 4, 6, 8, and 10 Gy), 15 min post substance addition. After the requisite incubation period (i.e., 24 or 48 h), cell survival was determined by the MTT test as previously described [17,18].

### 2.7. Irradiation

An Andrex SMART 200E (Yxlon International, Hamburg, Germany) X-ray producing equipment with the following characteristics was used: 200 kV, 4.5 mA, filtration of 2.5 mm of Al, and dose rate of 1.3 cGy/s at a focus object distance (FOD) of 35 cm. The experiments were performed at room temperature. For the determination of in vitro genotoxicity, whole human blood samples were exposed to 2 Gy X-rays at an FOD of 35 cm; while for the in vivo study, conscious and immobilized animals were whole body irradiated to a dose of 500 mGy at an FOD of 74 cm. For the determination of the radioprotective capacity, cell cultures grown in microplates were irradiated to different doses of X-rays (0, 4, 6, 8, and 10 Gy) at an FOD of 35 cm. At all times, the doses of radiation administered were continuously monitored inside the X-ray cabin by means of UNIDOS^®^ Universal Dosimeter with PTW Farme^®^ ionization chambers TW30010 (PTW-Freiburg, Freiburg, Germany), and the final radiation dose was confirmed by means of thermoluminescent dosimeters (TLDs) (GR-200^®^; Conqueror Electronics Technology Co Ltd., Beijing, China).

### 2.8. Statistical Analysis

Two different analysis were performed in this study. In the genoprotective study, analysis of variance complemented by a contrast of means to determine the degree of dependence and correlation between the variables was performed and further complemented with regression and linear correlation analysis amongst the quantitative variables. In the radioprotective study, the percentages of the surviving cells in the presence of the different substances were compared using analysis of variance (ANOVA) of repeated means, complemented by a least significant differences analysis to contrast pairs and means. *p* values of less than 0.01 (*p* < 0.01) were deemed significant.

In addition, in the genotoxicity analysis, we used the formula described by Sarma and Kesavan [19] to evaluate the protection factor (PF) regarding the reduction of the frequency of occurrence of MN:PF (%) = (Fcontrol – Ftreated/Fcontrol) × 100, where Fcontrol is the frequency of micronuclei in the irradiated control samples, and Ftreated is the frequency of micronuclei in the treated and irradiated samples. In the radioprotection analysis we modified this formula to adapt it to the cell survival cultures exposed to 10 Gy and incubated over a period of 48 h: PF (%) = (Mcontrol − Mtreated/Mcontrol) × 100, where Mcontrol is the mortality of the irradiated control cells, and Mtreated is the mortality of the cells treated with each substance and irradiated.

Finally, the dose reduction factor (DRF) was calculated as a ratio of radiation dose required to produce the same biological effect in the presence and absence of the radioprotector as described by Hall [16].

## 3. Results

### 3.1. Identification and Quantification of the Main Active Compounds in P. angolensis Seeds

The chromatogram produced according to HPLC analytical assay of *P. angolensis* seeds pointed to the presence of some plastoquinones/ubiquinones. Figure 1 shows the chromatogram of material obtained from *P. angolensis* seeds. The composition of the extracts in the chromatograms was assessed by comparing their relative retention times and UV spectra, which are a function of their molecular structures, and subsequently confirmed by mass spectrometry (HPLC-MS). The proposed compounds were the plastoquinones/ubiquinones sargahydroquinoic acid, (peak 1), sargaquinoic (peak 2), and sargachromenol (peak 3).

To confirm the structure of the above-mentioned compounds, the HPLC mass spectra data (Materials and Methods 2.4.2) of the three main peaks present in *Pycnanthus angolensis* seed extract were evaluated (Table 1).

Based on the results obtained, we propose that the peaks obtained in the HPLC-MS correspond to the potential generation of two adducts, one of which probably is due to Na+ binding [M + Na^+^] to the primary carboxyl group in the plastoquinone/ubiquinone molecule/skeleton and the other on the sterically less hindered hydroxyl group on the phenol ring of the type [M + 2Na-H]^+^. Thus, the case of sargahydroquinoic acid specifically has

426 (theoretical molecular weight) + (1 × 23) = 449 (signal shown);

426 (theoretical molecular weight) + (2 × 23 -1H) + = 471 (signal appearing).

Based on the deductions made from these results, it may be reasonable and consistent to imagine that the molecular structure of the main actives (peaks 1, 2 and 3) present in the plant material used in the study are compatible with sargahydroquinoic acid, (peak 1), sargaquinoic (peak 2), and sargachromenol (peak 3). Figure 2 shows the chemical structures of these compounds. Quantitative analysis of these compounds shows the following amounts (% weight) in the crushed seeds: sargahydroquinoic acid, 7.48%; sargaquinoic, 0.41% (peak 2); and sargachromenol, 1.42%.

### 3.2. Quantification of the Main Active Compounds in P. angolensis Seeds Extract (PASE)

Figure 3 shows the characteristic chromatogram of PASE. This HPLC chromatogram was characterized by a selective and significant increase in the sargahydroquinoic acid level. The contents (%weight) of the three main identified compounds in the ether extract of the crushed seeds were: sargahydroquinoic acid, 40.14% (peak 1); sargaquinoic, 0.35% (peak 2); and sargachromenol, 0.56% (peak 3). This extract was used in all the assays.

### 3.3. HPLC Analysis of Used Flavonoids, Polyphenols, and Plant Extracts

HPLC analysis of polyphenol distribution in the different extracts used in this study is as shown in Table 2. Diosmin is a very well-known flavonoid, specifically a flavone compound (C2=C3 double bond on flavonoid skeleton), widely used in the pharmaceutical field as peripheral vasoprotective agent. Quercetin is another flavonoid from the flavonol family (C2=C3 double bond and 3-OH radical), also widely used as reference compound in flavonoid research and a pharmaceutical raw material. Eriodictyol is a flavanone aglycone with the same substitution pattern as quercetin but without a C2=C3 double bond and 3-OH radical group. Apigenin is also a flavone compound with very significant anti-inflammatory properties. Green tea extract is one of the most popular extracts used as nutritional supplement for several health applications, with flavan-3-ol family of compounds (also named “catechins”) being the main flavonoid compounds present. Carnosic acid is the main diterpene from rosemary leaf extract, widely used as lipid antioxidant in food applications. Rosmarinic acid, a polyphenol with the “caffeoyl” structure, is widely present in the plant kingdom with interesting antioxidant and photoprotective properties. The chemical structures of the main flavonoids and polyphenols in the different extracts used in this study and the chemical structures of sulfur-containing compounds (amifostin and DMSO) are shown in Figure 3.

### 3.4. X-ray Genoprotective Effects: Antimutagenic Activity

No significant differences were determined between the frequency of occurrence of MN in human lymphocytes treated with the different substances tested and control lymphocytes, indicating the absence of genotoxicity effects of these substances. X-ray exposure produced a significant increase in MN frequencies in irradiated human peripheral blood lymphocytes (26 ± 2.1 MN/500 CB) when compared to the baseline micronuclei frequency portrayed in nonirradiated blood samples (10 ± 1.1 MN/500 BC) (*p* < 0.001), showing a genotoxic capacity of the 2 Gy of X-rays administered. Figure 4a shows the influence of timing of sample treatment (i.e., addition before and after X-ray exposure) on the frequency of MN in irradiated human lymphocytes. There was an observed decrease in the frequency of MN produced when the different test substances were administered, revealing the individual antigenotoxic capacity of each substance tested. It was further observed that the induced MN frequency was strongly influenced by the sequence and timing of the two treatment modalities. When human lymphocytes were treated with test substances before exposure to X-rays, the frequency of MN showed the following order with respect to irradiated control samples: RA < CA = API = T < D < AMF = C < PASE < Te (*p* < 0.001), where CA, API, and T did not show statistically significant differences between them (CA = API = T), and AMF and C also showed no significant differences between them (AMF = C). Finally, DMSO also showed a smaller reduction with respect to the frequency of occurrence of MN in the irradiated control samples (*p* < 0.01). However, unlike all previous substances, E and Q showed an increase in the frequency of MN with respect to irradiated controls (*p* < 0.001) that could be interpreted as an increase in the radiosensitivity of the samples treated with these substances.

However, when the different substances were administered after X-ray exposure, the MN frequencies were higher than what was observed in the pre X-ray treatments. It is clear that while CA showed significant antimutagenic activity, RA demonstrated a low degree of genoprotective activity, and DMSO along with AMF (sulfur-containing compounds) lost their genoprotective capacities against X-rays. The order of genoprotection from lowest to highest level of radiation induced MN frequency was CA < API < PASE = T < D < C < Te = RA < AM = DMSO < E < Q (*p* < 0.001). Figure 4b shows the protection factors of each treatment and how they varied according to the time of administration (before and after exposure to ionizing radiation).

Significant differences were not established between the frequency of MNPCEs/1000 PCEs in the animals treated with the different substances and the control animals, and no significant changes in the P/N and P/E ratios in any of the exposed animals compared with the control groups were observed, indicating the absence of genotoxicity of these substances. Exposure to X-rays produced a significant increase in MNPCEs/1000 PCEs (18.7 ± 1.1 MNPCEs/1000 PC) with respect to the baseline frequency presented by control animals (3.1 ± 1.1 MNPCEs/1000 PC) (*p* < 0.001), showing a genotoxic capacity of the 500 mGy of X-rays administered. Figure 5a shows the influence of timing of treatments before and after exposure to X-rays on in vivo MN frequency induced in polychromatic erythrocytes (PCEs) of mouse bone marrow in vivo, which also permits a comparison of the potential genotoxicity of X-rays irradiated control, versus the antimutagenic capacities of the different substances assayed. The frequency of induced micronuclei varied with the time of administration of the test substances and exposure to radiation. When the test substances were administered prior to irradiation, the radiation-induced MN frequencies (MNPCEs) ordered from lowest to highest was RA < CA < API < PASE = D < AMF = Te = DMSO (*p* < 0.001). However, when the different substances were administered after X-ray irradiation, the induced MN frequencies (MNPCEs) were higher than that observed in the groups that received treatment before X-ray. It is clear that while CA showed significant antimutagenic activity, RA demonstrated a low degree of genoprotective activity, while DMSO and AMF (sulfur-containing compounds) lost their genoprotective capacities against X-rays. The genoprotective capacity of these substances were as follows: CA < API < PASE < D = Te < RA < AMF < DMSO (*p* < 0.001).

Figure 5b shows the protection capacities of the substances tested, the order of efficacies being RA > CA > API > PASE = D > AMF = Te = DMSO for treatments before X-ray exposure and CA > AP > PASE > D = Te > RA > AMF > DMSO for treatments after X-irradiation. Differences in effects between CA, RA, and PASE in relation to the timing of substance administration (i.e., before or after exposure to radiation) can also be appreciated.

### 3.5. Radioprotective Effects against X-rays: Growth Inhibition

The concentrations of the tested substances that we selected for this study had no effect on cell survival. All cultures treated with the test substances but not exposed to irradiation were found to have cell survivals within 100% ± 5% for the different incubation periods. Moreover, no significant differences in cell survival induced by the treatments were established, demonstrating the absence of toxic effects of the substances administered. Figure 6 shows the percentage cell survival (%) of PNT2 and B16F10 melanoma cells assessed by the MTT cell viability test after administration of PASE during 24 and 48 h post irradiation incubation periods. In the irradiated PNT2 cells, PASE elicited an increase in cell survival at the highest radiation dose of 10 Gy used, which expressed the radioprotective capability of PASE for the two cell lines and at the two incubation periods studied (24 and 48 h) (*p* < 0.001). In the PNT2 cells, we established a protection factor of 35.5% and a dose reduction factor of 2.5 ± 0.2, respectively, after 48 h of incubation and exposure to 10 Gy of radiation, whereas in the B16F10 melanoma cells, a protection factor of 41.2% and DRF of 4 ± 0.2 were observed for the same incubation period (48 h).

Figure 7 shows an increase in the survival (%) of PNT2 cells after 24 and 48 h of incubation when treated with different substances (CA, API, RA, and PASE) and irradiated compared with cells that received only irradiation (irradiated controls), indicating the radioprotective capacities of these substances (*p* < 0.001).

## 4. Discussion

X-ray exposure produces a massive generation of reactive oxygen species (ROS)/free radicals in vivo. These ROS are formed by a sequential mechanism of electron transfer, through which molecular oxygen successively gives rise to a superoxide radical, hydrogen peroxide, and hydroxyl radical [12,20,21,22]. In general, ionizing radiation produces in the vicinity of DNA and its environs a large number of different radicals such as •OH, e^−^_aq_, and H• [7], which are mostly produced by the radiolysis water even in the absence of molecular oxygen. Its high reactivity produces an immediate reaction in the vicinity of its generation. However, when its generation is massive as a result of exposure to X-rays, the cytotoxic effect is no longer only local but can spread through reactive species and other radicals within the intracellular and even extracellular environment, increasing interaction with cellular phospholipids structures and inducing lipid peroxidation processes that increase the oxidative damage of DNA [20,21].

It is generally accepted that endogenously generated ROS and ROS that arise after exposure to X-rays are similar, but not necessarily identical. Both can be perpetuated through side reactions, for example, they can react with polyunsaturated fatty acids, which can cause tertiary biochemical reactions. However, the particular difference between metabolic ROS and those induced by ionizing radiation are based on their compartmentalization and their rate of appearance. In mammalian cells, ROS occur steadily and abundantly at frequent and changing time intervals, partly the result of metabolic reactions. An antioxidant enzyme system maintains an intracellular ROS concentration within a physiological range. In X-ray exposure, the generation of these ROSs is massive and can exceed any intracellular protection mechanism, affecting any place in the cell, and with an intensity that depends on the dose rate of absorbed radiation and the linear transfer of energy of the ionizing radiation administered [23].

Several authors have shown that under conditions of intense oxidative stress, such as during exposure to X-rays, when the endogenous antioxidant systems may be insufficient or defective, exogenous supplements or agents with the ability react with and eliminate free radicals could be used, as long as they contribute with a high degree of stability of the new intermediate neoformed radicals [21]. However, although the genoprotective effects are mentioned as being scavenging of •OH radicals [10,11,12,13,14,19,23,24,25,26,27], new DNA protection mechanisms have been described in recent years: the displacement of water in the extended hydration shell of DNA, the energy loss of low-energy electrons due to the scattering at vibrational water modes [28,29], the resulting decrease in secondary structure, and the ability to protect cells from stress conditions and to prevent cell damage by maintaining an elevated level of the Hsp70 [30]. To quantify the relative contributions of these different protective mechanisms, further work is needed.

This study attempted to obtain an extract from a suitable part of African nutmeg *Pycnanthus angolensis* (PASE) as well as quantify its radioprotective capacity when administered before (pre) and immediately after (post) exposure to ionizing radiation in line with results of our previous studies. In those studies, different test substances were evaluated for radioprotection, some of which expressed higher degrees of protection against harmful damage induced by ionizing radiation when compared with reference radioprotective compounds. In order to achieve these objectives and to enable effective comparisons, the same experimental protocols used in those previous studies were adapted for this study [8,9,17,18] but used a different ionizing agent, X-rays, which also conditioned new exposure times and dose rate.

We have previously described the antimutagenic capacity of some of the substances used in this study as comparative controls (RA, CA, API, D) against genotoxic damage induced by gamma radiation [8,10,11,27]. In this study, when these substances are administered before or after exposure to X-rays, they have lower anitmutagenic capacities that show no statistically significant differences. According to previous authors, our results show that when ionizing radiation has similar linear energy transfers, it conditions a similar relative biological effectiveness; therefore, the intensity of the radio-induced damage is similar [7,16]. This would explain that the genoprotective capacity of some substances tested in this study are similar to X-rays as well as gamma radiation [8].

We have not found specific information on the cellular uptake of the different compounds in relation to the micronucleus or MTT assays. We have previously shown that the reduction of micronuclei induced by ionizing radiation in biological systems cannot be directly ascribed to one single compound with a peculiar chemical structure, although the observed genoprotective effects induced by the presence of some compounds seem to be related to their antioxidant capacities and bioavailability in the cellular milieu [8]. Consequently, we observed that flavan-3-ols showed the greatest protective capacity of all polyphenols evaluated in our previous study [12], while other flavonoids known to have high antineoplastic and antiproliferative capacities showed lower antimutagenic capacities [31,32]. This may explain the increase in micronuclei formation obtained after treatments with Q and E since both present a flavonoid structure with a catechol group in the B ring that gives them pro-oxidant capacities, even at the low concentrations used in this study. When reacted with the superoxide radical, this can lead to the generation of hydrogen peroxide, thus increasing its cytotoxic capacity [33,34,35,36].

In addition, this genoprotective capacity was also found to depend on the degree of polymerization and solubility of the substances assessed, since both modify their bioavailability [8,10,11]. We believe that all of the above are equally relevant to PASE, which we used as a genoprotective substance against IR-induced damage in this study.

We have not found references on antigenotoxic activity of *Pycnanthus angolensis.* However, its genoprotective capacity is greater than that shown by the AMF, the only radioprotector used in radiation oncology. In this sense, when an extract of PASE is administered before exposure to IR, it is observed to have a medium radioprotective capacity similar to that of AMF; however, it is found to be superior to AMF when administered after exposure to X-rays, an observation which could be explained by the increased activity in the lipid peroxidation process [8,9].

We have not found references on the radioprotective activity of *Pycnanthus angolensis*. Our results on cell survival, protection factor (PF), and dose reduction factor (DRL) also confirm the radioprotective effect of PASE against cytotoxic damage induced by IR on normal prostate (traditionally considered as radiosensitive cells) and melanoma tumor cells (considered as radioresistant cells). This radioprotective effect is less intense than determined for RA and CA for PNT2 cells, but it did not portray the paradoxical radiosensitizing effect of these substances that we previously described in melanoma cells treated with RA and CA. In B16F10 melanoma cells irradiated with X-rays, both substances (RA and CA) are shown as potent radiosensitizing agents to reduce cell survival, suggesting a mechanism of activation of pheomelanin production that would consume intracellular glutathione causing the decrease of endogenous protection mechanisms [17,18].

Regardless of its applications in traditional medicine, components of PASE are habitually consumed in human diets and have been included in dietary supplements for decades within certain concentration ranges without reported toxicity. With the exception of AMF, which is known to have a high degree of toxicity, the substances analyzed in this study are also common components of human diets [22,25]. Evidently, before administering any compounds to humans in an ionizing radiation exposure scenario, studies should be performed on a wider range of cell lines, especially since different types of tissues can act quite diversely to the same compound with/without radiation exposure. Further, the effects of antioxidant supplements in oncology may be harmful. Although some studies have suggested that antioxidants can protect normal tissues from chemotherapy- or radiation-induced damage, others have claimed that supplementary antioxidants during chemotherapy and radiation therapy should be discouraged because they may actually protect the tumor cells and so reduce survival of the patient [8].

In this regard, it is worth pointing out two scenarios in which a dietary supplement containing these substances, specifically AR and PASE, may have utility in the reduction of the stochastic effects induced by ionizing radiation: the protection of workers professionally exposed to ionizing radiation and patients undergoing medical radiodiagnostic examinations. In this way, it offers a possibility of increasing the levels of endogenous protection against damage induced by ionizing radiation in professionally exposed workers by scavenging the free radicals produced and reducing the harmful effects of accidental exposures; while patients who consume these substances prior to performing radiological examinations would be endowed with the capacity to reduce or eliminate a possible stochastic effect induced by IR. PASE offers an improved protective capability over RA as it offers the possibility of protecting cells against damage even after exposure to ionizing radiation, a property which could be exploited to mitigate the harmful effects of ionizing radiation during accidental or emergency situations in which workers are exposed to radiation without prior knowledge. In conclusion, given the possibility of stochastic effects, the use of innocuous substances, such as PASE, may offer protection against biological damage induced by ionizing radiation by augmenting the endogenous antioxidant protection mechanisms of the individual.

## Figures and Tables

**Figure 1 jcm-09-00006-f001:**
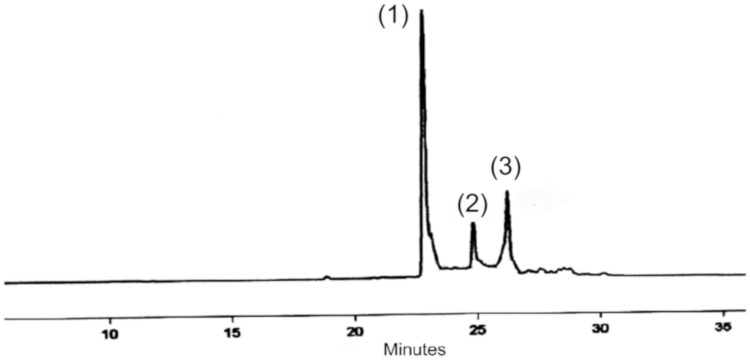
Characteristic chromatogram (Materials and Methods 2.4.1) of *Pycnanthus angolensis* seed extract (PASE), monitored at 250 nm. Peaks: (1) Rt 22.7 min, sargahydroquinoic acid; (2) Rt 24.8 min, sargaquinoic acid; (3) Rt 26.4 min, sargachromenol.

**Figure 2 jcm-09-00006-f002:**
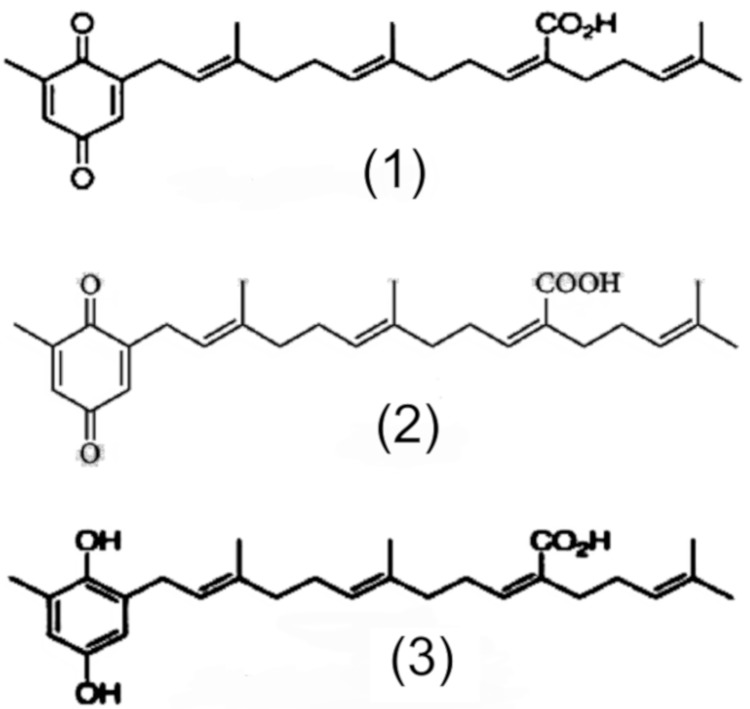
Possible chemical structures related to the results obtained from the PASE.

**Figure 3 jcm-09-00006-f003:**
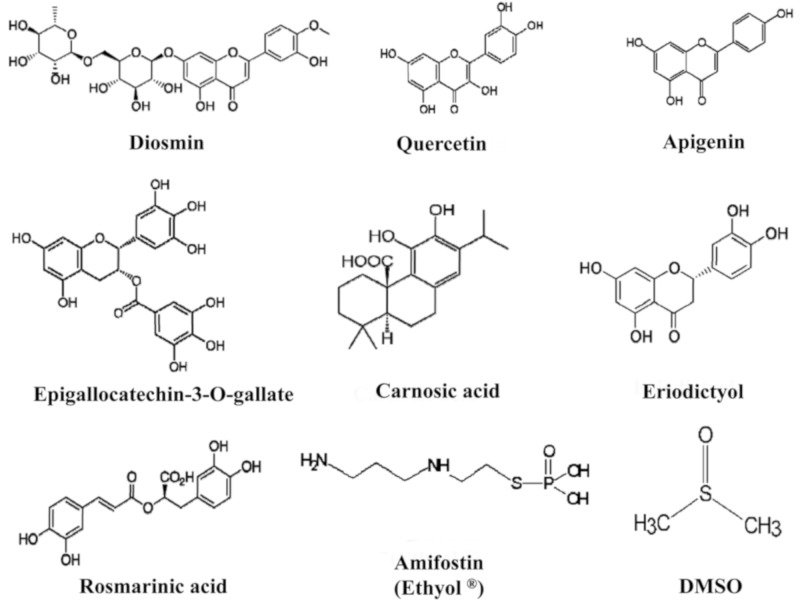
Chemical structures of different substances tested in this study.

**Figure 4 jcm-09-00006-f004:**
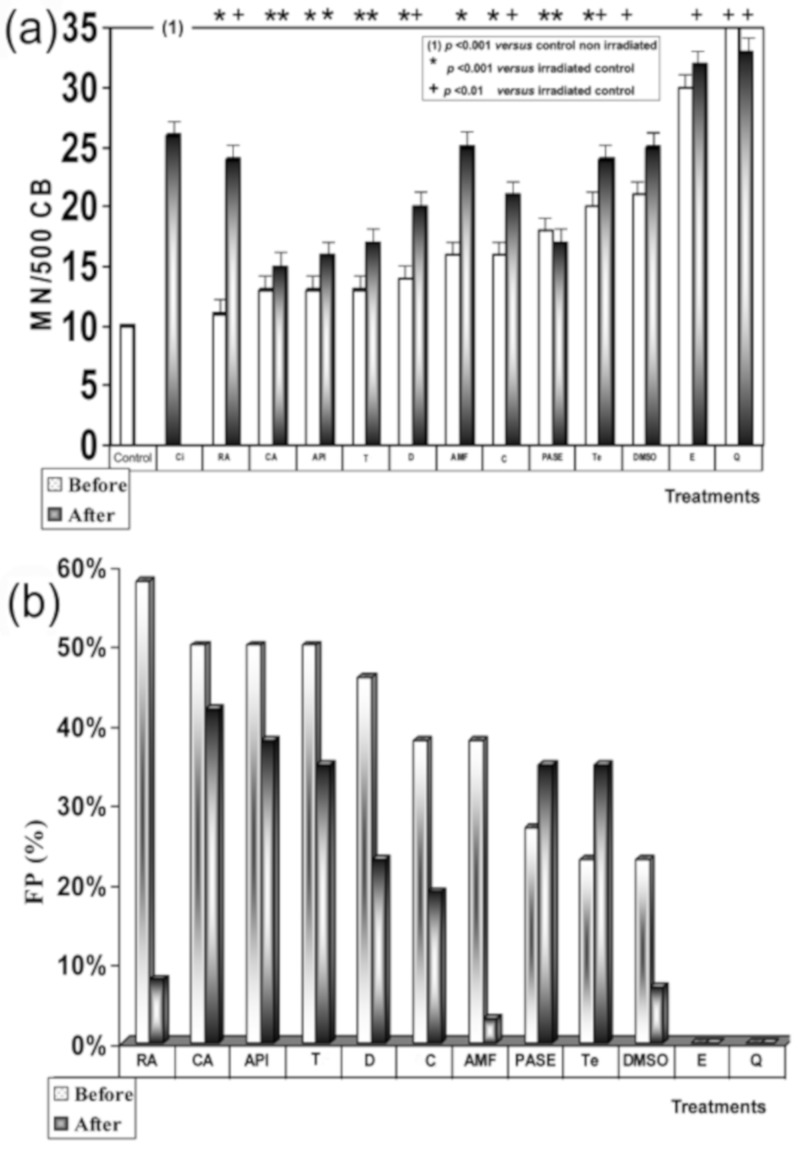
“In vitro” genoprotective effects against X-rays: (**a**) frequency of MN/500 CB in irradiated human lymphocytes blocked with cytochalasin B; (**b**) protection factor of PASE and of the other substances tested.

**Figure 5 jcm-09-00006-f005:**
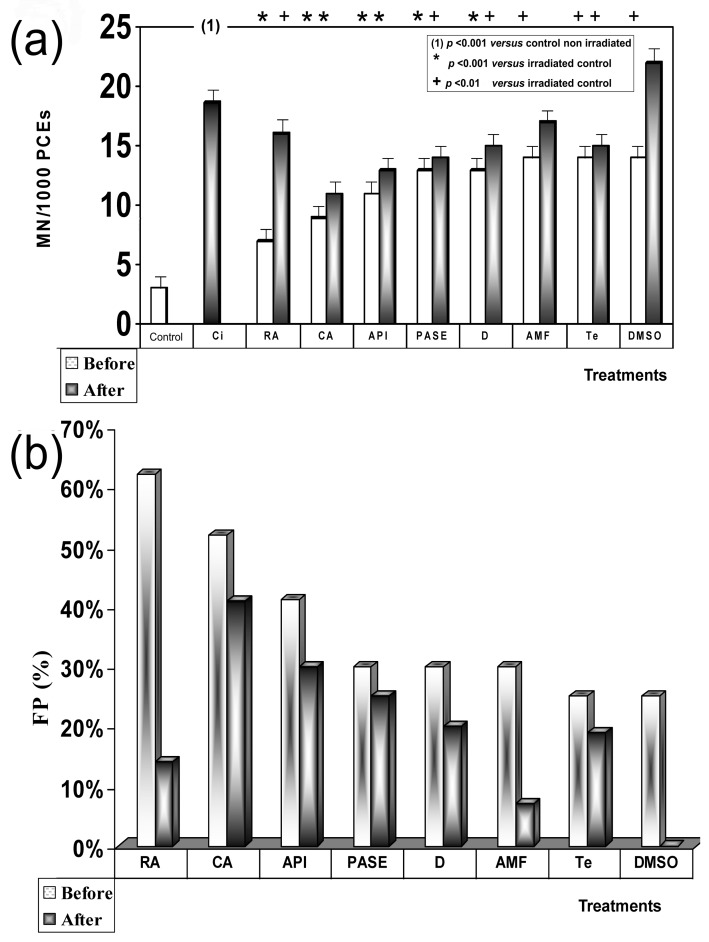
“In vivo” genoprotective effects against X-rays: (**a**) frequency of MN/1000 PCEs in mouse bone marrow; (**b**) protection factor of PASE and of the other substances tested. Radioprotective effects against X-rays. Protection capacities of the substances tested.

**Figure 6 jcm-09-00006-f006:**
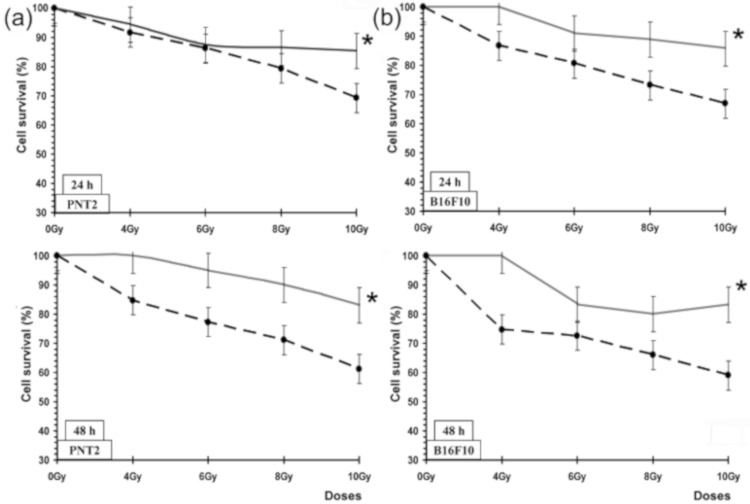
Cell survival (%) of PNT2 and B16F10 melanoma cells assessed by the MTT cell viability test after administration of PASE: (**a**) PNT2 cells after 24 h and 48 h incubation periods; (**b**) B16F10 melanoma cells after 24 h and 48 h incubation periods. * *p* < 0.001 versus control irradiated. Data are the mean ± standard error of eight independent experiments.

**Figure 7 jcm-09-00006-f007:**
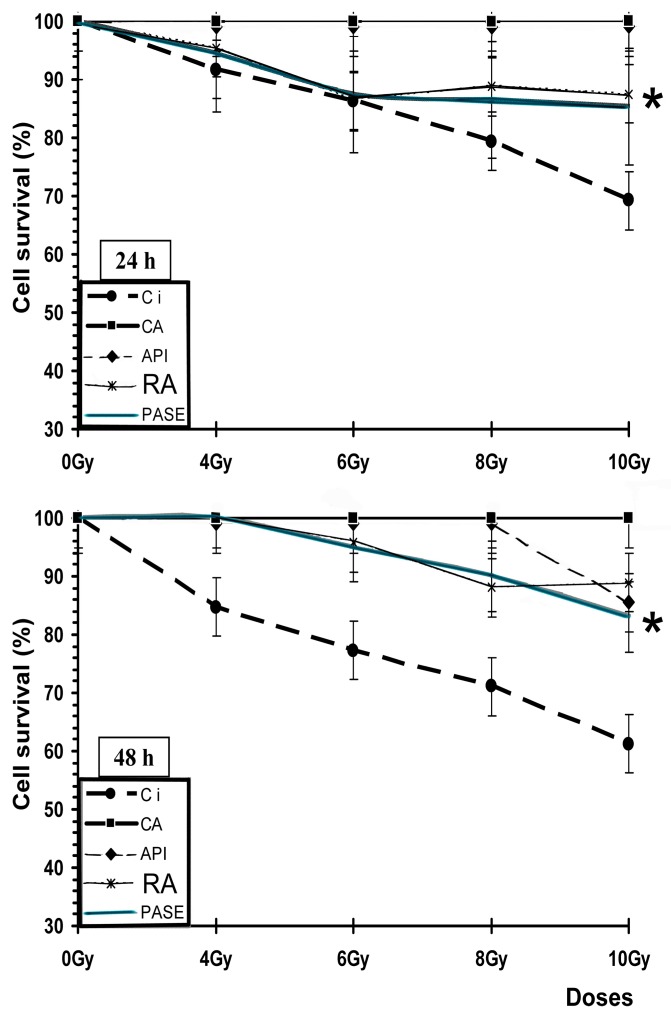
Cell survival (%) of PNT2 assessed by the MTT cell viability test after administration of different substances for 24 and 48 h incubation periods (Ci, irradiated control; CA, carnosic acid; API, apigenin; RA, rosmarinic acid). * *p* < 0.001 versus control irradiated. Data are the mean ± standard error of eight independent experiments.

**Table 1 jcm-09-00006-t001:** Experimental results (Materials and Methods 2.4.2) versus theoretical structural data of proposed compounds present in *Pycnanthus angolensis* seeds, monitored at 280 nm (HPLC-MS).

Parameter	Peak 1	Peak 2	Peak 3
Proposed structure	Sargahydroquinoic acid	Sargaquinoic acid	Sargachromenol
Molecular weight of proposed compound	426	424	424
Molecular formula of proposed compound	C_27_H_38_O_4_	C_27_H_36_O_4_	C_27_H_36_O_4_
Retention time in HPLC-MS analysis	27.4 min	29.3 min	31.6 min
M + H^+^	449	447	447
M + Na	471	469	469
Molecular weight obtained from HPLC-MS	448	446	446
Theoretical and experimental mass difference	+ 22	+ 22	+ 22

**Table 2 jcm-09-00006-t002:** Distribution of chemical structures by HPLC analysis in the different extracts used in this study.

Used Compound Extracts	Chemical Structure	Main Compounds	Content (%) ^1^
Diosmin	Flavone	Diosmin	93.44
	Flavanone	Hesperidin	1.78
	Flavone	Isorhoifolin	0.23
	Flavone	Diosmetin	0.18
		Other flavonoids	0.43
Quercetin	Flavonol	Quercetin	94.37
	Flavonol	Isoquercitrin	1.12
	Flavonol	Rutin	0.75
		Other flavonoids	0.56
Apigenin	Flavone	Apigenin	95.72
	Flavone	Rhoifolin	1.38
	Flavanone	Naringenin	0.64
		Other flavonoids	0.67
Eriodictyol	Flavanone	Eriodictyol	93.12
	Flavanone	Eriocitrin	2.56
	Flavanone	Hesperidin	0.52
		Other flavonoids	0.71
Green Tea Extract	Flavan-3-ol	Epigallocatechin 3-O-gallate	57.89
	Flavan-3-ol	Epigallocatechin	14.65
	Flavan-3-ol	Epicatechin 3-O-gallate	6.71
	Flavan-3-ol	Epicatechin	6.11
Carnosic acid	Diterpene	Carnosic acid	76.44
	Diterpene	Carnosol	4.68
	Diterpene	12-methyl-carnosic acid	3.70
		Other diterpenes	1.23
Rosmarinic acid	Di-caffeoyl compound	Rosmarinic acid	94.22
	Caffeoyl co.	Di-hydroxy-cinnamic acid	0.65
		Other polyphenols	1.78

^1^ Absolute value as is.

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
