# Peer review of "Radioprotective and Antimutagenic Effects of Pycnanthus angolensis Warb Seed Extract against Damage Induced by X rays"

_jcm, 2019, doi:10.3390/jcm9010006_

Round 1

Reviewer 1 Report

The authors investigate the radioprotective and antimutagenic effects of seed extracts from P. angolensis seed extracts (PASE) against xray induced damage.
They extracted, purified and analyzed the PASE. Xray irradiation with doses up to 10Gy were performed for various samples under the presence of different compounds. The samples were mouse bone marrow (in vivo), human lymphocites (in vitro), and two cell lines which are eitehr knwon to be radioresistant (B16F10) or radiosensitive (PNT2).
The radiation damage and effects of the compounds were analysed by micronucleus assays and cell viability assays.
The results show different protective effects of the tested compounds.
The authors recommand probable applications of PASE in occupational and the medical field.

General remarks:

In general the manuscript is of interest to the reader and well written. Nevertheless, before recommanding publication I would suggest the authors to extend and improve the manuscript on some points.
Especially in the discussion I found some critical points where, in my opionion, improvement is needed. The discussion should be more detailed and some statements should be redefined or clarified to not mislead the reader.
This is especially important since possible applications point toward the use of the substances involving clinical and occupational exposure situations of human beings.
Therfore statements and recommendations should be made very careful with regard to the usage of the substances in humans before I can recommend punlication.
Taking this and the points listed below into account I would recommand a major revision.

The main criticism are overgeneralized statements in the discussion regarding possible applications to humans. They should be carefully revised.
These points and other comments are discussed in the following.

Introduction:
-In the introduction it would be appropriate to cite the standard textbook "Free-Radical-Induced DNA Damage and Its Repair" by "von Sonntag" (https://www.springer.com/gp/book/9783540261209) which might be as well helpful for extending the discussion about antioxidant properties and interpreting the data obtained.

Methods:
-the desciption of sample treatment with administering the compounds before and after is a little ambigous (line 158). it should be clearly stated whether the compounds were either added (before OR after) or added (before OR before and after). I assume the former case, but the text might suggest the latter as well, so this crucial point should be reformulated to have no doubts.
-MTT test: is there any information of the cellular uptake of the different compounds?
this is very important to access protective methods (compare Rieckmann et al https://www.nature.com/articles/s41598-019-43040-w)
-Line 222 "significant p values ... were deemed significant" should be reformulated to
-all equations thoughout the text: please reformat them, to make them readable. use short simbols and define their meaning in the text. Please use "some" formula editor, otherwise they are easily misunderstood. (e.g. line 226 etc.)

Results:
- figure 6 clearly describe which curve is which. what is the number of replicates etc?
-figure 1: please show both, HPLC and MS results to make assignments and interpretation more understandable to the interested reader
-figure 1/tab.1: clearly assign the peaks with refrence to tab 1
-Ihe effects of the different compounds on irradiation outcome was given a couple of times (e.g. line 305) in terms of different signs (<,=). I I understand correctly the two outcomes stronger "<" and similar "=" are related to a p<0.001 values when comparing the results of measurement outcome of different compounds. If so, please state it explicitely and add 1-2phrases to make it immeadiatly comprehensive to the reader which will help to avoid misunderstanding. If the assignment are intennded to be interpretet differently, please explain in detail.
-it might be of interest to include a table or figure with the data of reated/untreated cells even tough there was no significant difference observed between treatments without irradiation.
-dose reduction factors should be given with uncertainity values

Discussion:
This part is the most critical one and needs careful revision.

- line 372: formulation is a little unclear: ROS, especially the most important one, OH radicals, are proudced after ionization of water and subsequent reactions (compare von sonntag).
-line 436: briefly explain the "paradoxial radiosensitizing effetcs" you mentioned to provide a more complete pciture to the reader/
-line 410: discuss possible reasons for lower antimutagenic capacities. providing that a publication of the article in higher if jounral might be considered to reach a wider audience.
-line 411: that radiation of same LET causes similar amount of biological damage is not really surprising (compare work of Kellerer "Fundamentals of Microdosimetry" 1985). since this study only includes one xray energy to state the this study confirm something about the LET concept is a little bold. I would recommand reformulating this phrase slightly.
-line 432: add that its an extract.
-The protective affects are mentioned as being scavenging of OH radicals. I agree that this is most likely. But a proof would be very beneficial or should be at least discussed. To access scavenging capabilities of such natural compounds compare for example Hahn et al (https://pubs.rsc.org/en/content/articlehtml/2017/cp/c7cp02860a).
Since damage to DNA is most detrimental to cells, scavengign would be most effective ocurring near the nucleus. This is not possible when there is no uptake of the compounds. So uptake of the compounds should be tested (compare Buommini 2005 http://www.bioone.org/doi/abs/10.1379/CSC-101R.1).
-How does the interpretation as "protection by OH scavenging" can be explained when DMSO is much worse, which is known as a very effective OH scavenger? Please discuss this point.
-Before administering any compounds to humans in an exposure scenarion to ionizing radiation, studies should be performed on a wider range of cell lines. Especially since different types of tissues can act quite diversly to the same compound with/without radaition exposure.
-Applications in medical diagnostics: As desribed above, different types of risks should always be weightened against each other. Do not overgeneralize statements. The matter is, due to its nature, quite complex.
-When it is being written about certain concentration ranges (line 440) they should be given (with reference to the literature) and compared to the concentrations in this study.
-Point clearly out, that prior to any application during radiation therapy treatment further studies are needed. They are needed to make sure that the respective cancerous tissues do not benefit "more" from the protective effects than the healthy tissue(compare Rieckmann et al https://www.nature.com/articles/s41598-019-43040-w).
-The final statement of the discussion, that the compounds should be used even "against any dose of radiation however small they may be (line 457)" is to be seen very critical, and should be removed. The possible advertise effects of administration of all kind of drugs should be always weightened carefully against low-dose effects of radiation.

Author Response

Reviewer #1: 

The authors investigate the radioprotective and antimutagenic effects of seed extracts from P. angolensis seed extracts (PASE) against xray induced damage.

They extracted, purified and analyzed the PASE. X-ray irradiation with doses up to 10 y were performed for various samples under the presence of different compounds. The samples were mouse bone marrow (in vivo), human lymphocytes (in vitro), and two cell lines which are either known to be radioresistant (B16F10) or radiosensitive (PNT2).

The radiation damage and effects of the compounds were analysed by micronucleus assays and cell viability assays.

The results show different protective effects of the tested compounds.
The authors recommend probable applications of PASE in occupational and the medical field.

General remarks:

In general, the manuscript is of interest to the reader and well written. Nevertheless, before recommending publication I would suggest the authors to extend and improve the manuscript on some points.

Especially in the discussion I found some critical points where, in my opinion, improvement is needed. The discussion should be more detailed and some statements should be redefined or clarified to not mislead the reader.

This is especially important since possible applications point toward the use of the substances involving clinical and occupational exposure situations of human beings.

Therefore, statements and recommendations should be made very careful with regard to the usage of the substances in humans before I can recommend publication.

Taking this and the points listed below into account I would recommend a major revision.

The main criticism are overgeneralized statements in the discussion regarding possible applications to humans. They should be carefully revised.

These points and other comments are discussed in the following.

Comment: Introduction:

-In the introduction it would be appropriate to cite the standard textbook "Free-Radical-Induced DNA Damage and Its Repair" by "von Sonntag" (https://www.springer.com/gp/book/9783540261209) which might be as well helpful for extending the discussion about antioxidant properties and interpreting the data obtained.

Response. We have included the “von Sonntag” reference and incorporated it in the Introduction (line 55). Furthermore, we have quoted it again in our discussion below.

Comment: Methods:

-the description of sample treatment with administering the compounds before and after is a little ambigous (line 158). it should be clearly stated whether the compounds were either added (before OR after) or added (before OR before and after). I assume the former case, but the text might suggest the latter as well, so this crucial point should be reformulated to have no doubts.

Response. The reviewer's interpretation is appropriate. We have reformulated the paragraph to avoid doubts. The paragraph now reads (line 156):

Twenty microliters (20 µl) of a 20 µM concentration of the test substances were added to 2 ml of the heparinized human blood samples at two different times: immediately before exposure to X-rays (treatment before irradiation) or immediately after exposure to the X-rays (post irradiation treatment).

Comment:-MTT test: is there any information of the cellular uptake of the different compounds? this is very important to access protective methods (compare Rieckmann et al https://www.nature.com/articles/s41598-019-43040-w)-

Response. We are not privy to this type of information regarding the substances tested in relation to MTT.

Comment:-Line 222 "significant p values ... were deemed significant" should be reformulated to -all equations thoughout the text: please reformat them, to make them readable. use short simbols and define their meaning in the text. Please use "some" formula editor, otherwise they are easily misunderstood. (e.g. line 226 etc.)

Response. We have modified the paragraph and reformulated the equations. Previously, the sentence read as follows:In addition, in the genotoxicity analysis, we used the formula described by Sarma and Kesavan [17] to evaluate the protection factor (PF) regarding the reduction of the frequency of occurrence of MN: PF (%) = (Frequency MNcontrol irradiated- Frequency MNtreated with substances and irradiated/ Frequency MNcontrol irradiated) x 100. In the radioprotection analysis we modified this formula to adapt it to the cell survival cultures exposed to 10 Gy and incubated over a period of 48h: PF (%) = (Mortality of the irradiated control cells - Mortality of the cells treated with test substances and irradiated/ Mortality of the irradiated control cells) x 100.”

This has been modified and the paragraph now reads (line 226):In addition, in the genotoxicity analysis, we used the formula described by Sarma and Kesavan [19] to evaluate the protection factor (PF) regarding the reduction of the frequency of occurrence of MN: PF(%)=(Fcontrol -Ftreated/Fcontrol) x 100, where Fcontrol is the frequency of micronuclei in the irradiated control samples and Ftreated is the frequency of micronuclei in the treated and irradiated samples.  In the radioprotection analysis we modified this formula to adapt it to the cell survival cultures exposed to 10 Gy and incubated over a period of 48h: PF(%) =(Mcontrol - Mtreated / Mcontrol) x 100, where Mcontrol is the mortality of the irradiated control cells and Mtreated is the mortality of the cells treated with each substance and irradiated”.

Comment: Results:

- figure 6 clearly describe which curve is which. what is the number of replicates etc?

Response. We have incorporated in the figure 6 the description of each curve. The legend of the figure has also been modified to reflecte the changes made. The modified figure is depicted as follows:

(Please see the attachment: Figure 6)

Comment:-figure 1: please show both, HPLC and MS results to make assignments and interpretation more understandable to the interested reader

-figure 1/tab.1: clearly assign the peaks with refrence to tab 1

Response. The location of Figure 1 misled the interpretation of the results. It has now been properly placed in the text. We performed two different chromatographic analyzes that are described in the sections 2.4.1 and 2.4.2 of material and methods. respectively.

The first chromatographic analysis to perform an initial elucidation of active PASE compounds is described in Material & Methods 2.4.1. This first chromatogram corresponds to Figure 1, whose legend we have modified. The paragraph now reads: Figure 1. Characteristic chromatogram (material and methods 2.4.1) of Pycnanthus angolensis Seed extract (PASE), monitored at 250 nm. (Peaks: (1) Rt 22.7 min, sargahydroquinoic acid; (2) Rt 24.8 min, sargaquinoic acid; (3) Rt 26.4 min, sargachromenol.)”

Subsequently, we performed a second chromatographic analysis to perform the determination and identification of assets by HPLC-MS, under the conditions described in section 2.4.2 of material and methods. These new chromatographic conditions (basic change of column and mobile phase) were necessary to obtain better results in this analytical determination. From this second chromatogram we have considered more clarifying to make a Table (Table 1) with the experimental results obtained from the PASE.

We have modified the legend of Table1, now reads:” Table 1. Experimental results (material and methods 2.4.2) versus theoretical structural data of proposed compounds present in Pycnanthus angolensis Seed, monitored at 280 nm HPLC-MS.”

This chromatogram has allowed us to confirm the experimental results obtained by the first chromatographic analysis, but it is important to consider that the retention times (Rt) of the three isolated and identified compounds are different in both chromatographic profiles.

The chromatogram obtained is described in Table 1 is as follows:

(Please see the attachment:Table 1 )

With this background we have modified Figure 1 and the text of Figure 1. Now it shows:

(Please see the attachment: Figure 1.)

Comment:-The effects of the different compounds on irradiation outcome was given a couple of times (e.g. line 305) in terms of different signs (<,=). I I understand correctly the two outcomes stronger "<" and similar "=" are related to a p<0.001 values when comparing the results of measurement outcome of different compounds. If so, please state it explicitly and add 1-2phrases to make it immediately comprehensive to the reader which will help to avoid misunderstanding. If the assignment are intended to be interpreted differently, please explain in detail.

Response. Indeed, the initial interpretation of the reviewer is appropriate. To improve understanding, we have modified the paragraph. It was read earlier (line 305) as: When human lymphocytes were treated with the test substances before exposure to X-rays, the frequency of MN induced by irradiation followed the order: RA<CA=API=T<D<AMF=C<PASE<Te<DMSO<E

This has been modified and the paragraph now reads (line 309) as: “When human lymphocytes were treated with test substances before exposure to X-rays, the frequency of MN shows the following order with respect to irradiated control samples: RA<CA=API=T<D<AMF=C<PASE

Comment:-it might be of interest to include a table or figure with the data of reated/untreated cells even tough there was no significant difference observed between treatments without irradiation.

Response. The concentrations of the tested substances that we have selected for this study have no effect on the cell survival. All cultures treated and non-irradiated were found within 100 ± 5% cell survival during the different incubation periods and no significant differences induced by the treatments have determined. Therefore, we had no interest in incorporation this information into the manuscript.

Comment: -dose reduction factors should be given with uncertainity values

Response. We have incorporated these values. Now the paragraph reads. (line 367): In the PNT2 cells, we established a protection factor of 35.5% and a dose reduction factor of 2.5±0.12 respectively after 48 h of incubation and exposure to 10 Gy of radiation, whereas in the B16F10 melanoma cells a protection factor of 41.2% and DRF of 4±0.2 were observed for the same incubation period (48 h)”.

Comment:Discussion:
This part is the most critical one and needs careful revision.

- line 372: formulation is a little unclear: ROS, especially the most important one, OH radicals, are proudced after ionization of water and subsequent reactions (compare von sonntag).

Response. We have modified that aspect of the paragraph, which now reads (line 385):

“X-ray exposure produces a massive generation of reactive oxygen species/free radicals (ROS) in vivo. These ROS are formed by a sequential mechanism of electron transfer, through which molecular oxygen successively gives rise to a superoxide radical, hydrogen peroxide and hydroxyl radical [12,20-22]. In general, ionizing radiation produces in DNA and its model systems a large number of different radicals through the action ofOH, eaq and H [7].  Its high reactivity produces an immediate reaction in the vicinity of its generation; However, when its generation is massive as a result of exposure to X-rays, the cytotoxic effect is no longer only local but can spread through reactive species and other radicals within the intracellular and even extracellular environment, increasing interaction with cellular phospholipids structures of and inducing lipid peroxidation processes that increase the oxidative damage of DNA [20,21]

Comment:-line 436: briefly explain the "paradoxial radiosensitizing effetcs" you mentioned to provide a more complete pciture to the reader/

Response. Previously that paragraph read:This radioprotective effect is less intense than determined for RA and CA for PNT2 cells but did not portray the paradoxical radiosensitizing effect of these substances that we previously described in melanoma cells treated with RA and CA [15,16].”

This has been modified and the paragraph now reads (line 455):

“This radioprotective effect is less intense than determined for RA and CA for PNT2 cells but did not portray the paradoxical radiosensitizing effect of these substances that we previously described in melanoma cells treated with RA and CA. In B16F10 melanoma cells irradiated with X-rays, both substances (RA and CA) are shown as potent radiosensitizing agents to reduce cell survival, suggesting a mechanism of activation of pheomelanin production that would consume intracellular glutathione causing the decrease of endogenous protection mechanisms [15,16]”.

Comment:-line 410: discuss possible reasons for lower antimutagenic capacities. providing that a publication of the article in higher if journal might be considered to reach a wider audience.

Response. Previously the paragraph read: “We have previously shown that the reduction of micronuclei induced by ionizing radiation in biological systems cannot be directly ascribed to one single compound with a peculiar chemical structure although observed genoprotective effects induced by the presence of some compounds seem to be related to their antioxidant capacities and bioavailability in the cellular milieu [7]. Consequently, we observed that flavan-3-ols showed the greatest protective capacity of all polyphenols evaluated in our previous study [11], while other flavonoids known to have high antineoplastic and antiproliferative capacities showed lower antimutagenic capacities [28,29]. This capacity was also found to depend on the degree of polymerization and solubility of the substances assessed, since both modify their bioavailability [7,9,10]. We believe that all of the above equally relevant to PASE which we used as a genoprotective substance against IR-induced damage in this study”.

This has been modified and the paragraph now reads as (line 435):

“We have previously shown that the reduction of micronuclei induced by ionizing radiation in biological systems cannot be directly ascribed to one single compound with a peculiar chemical structure although observed genoprotective effects induced by the presence of some compounds seem to be related to their antioxidant capacities and bioavailability in the cellular milieu [8]. Consequently, we observed that flavan-3-ols showed the greatest protective capacity of all polyphenols evaluated in our previous study [12], while other flavonoids known to have high antineoplastic and antiproliferative capacities showed lower antimutagenic capacities [31,32]. It is the situation of our results obtained in the treatments with Q and E since both present a flavonoid structure with a catechol group in the B ring that gives them a proxidant capacity, even at the low concentrations used in this study, and that when reacting with the radical superoxide can generate hydrogen peroxide, increasing its cytotoxic capacity [33-36].

In addition, this genoprotective capacity was also found to depend on the degree of polymerization and solubility of the substances assessed, since both modify their bioavailability [8,10,11]. We believe that all of the above are equally relevant to PASE which we used as a genoprotective substance against IR-induced damage in this study”

Comment: -line 411: that radiation of same LET causes similar amount of biological damage is not really surprising (compare work of Kellerer "Fundamentals of Microdosimetry" 1985). since this study only includes one x ray energy to state the this study confirm something about the LET concept is a little bold. I would recommand reformulating this phrase slightly.

Response. Previously the sentence read: Our results confirm that when ionizing radiation have similar Linear Energy Transfers, it conditions a similar Relative Biological Effectiveness, and therefore, the intensity of the radio-induced damage is similar [18]. All of this suggests, therefore, that the genoprotective capacity of these substances should be similar to either of both ionizing agents (X-rays and radiation gamma).”

This has been modified and the paragraph now reads (line 426):

 According to the previous authors our results show that when ionizing radiation have similar Linear Energy Transfers, it conditions a similar Relative Biological Effectiveness, and therefore, the intensity of the radio-induced damage is similar [7,16]. All this would explain, therefore, that the genoprotective capacity of some substances tested in this study be similar to either of both ionizing agents (X-rays and radiation gamma [8]).

Comment: -line 432: add that its an extract.

Response. This comment has been added. It previously read as:

”In this sense, when PASE is administered before exposure to IR it is observed to have a medium radioprotective capacity similar to that of AMF;”

It now reads: “In this sense, when an extract of PASE is administered before exposure to IR it is observed to have a medium radioprotective capacity similar to that of AMF;”

Comment:-The protective affects are mentioned as being scavenging of OH radicals. I agree that this is most likely. But a proof would be very beneficial or should be at least discussed. To access scavenging capabilities of such natural compounds compare for example Hahn et al (https://pubs.rsc.org/en/content/articlehtml/2017/cp/c7cp02860a).

Since damage to DNA is most detrimental to cells, scavengign would be most effective ocurring near the nucleus. This is not possible when there is no uptake of the compounds. So uptake of the compounds should be tested (compare Buommini 2005 http://www.bioone.org/doi/abs/10.1379/CSC-101R.1).

Response.  This has been modified and the paragraph now reads (line 410):  “However, although the genoprotective effects are mentioned as being scavenging of •OH radicals [10-14, 19, 23-27], new DNA protection mechanisms have been described in recent years: the displacement of water in the extended hydration shell of DNA, the energy-loss of Low Energy Electrons due to the scattering at vibrational water modes [28]), the resulting decrease in secondary structure and the ability to protect cells from stress conditions and to prevent cell damage by maintaining an elevated level of the Hsp70 [29]). To quantify the relative contributions of these different protective mechanisms further work is needed”.

Comment:-How does the interpretation as "protection by OH scavenging" can be explained when DMSO is much worse, which is known as a very effective OH scavenger? Please discuss this point.

Response. We have also studied the genoprotective capacities (figures 4 and 5) and radioprotective capacities of DMSO for two reasons: one, being an antioxidant substance with the characteristics of substance with sulfur; another, to be used in cell culture media usually. At high doses it has a slight genoprotective activity if it is administered before X-ray irradiation, but this genoprotection disappears if it is administered after exposure to X-rays, in a similar way to what we determine with the other sulfur substance (AMF) (lines 321 and 347).

In the radioprotective study, the final concentration of DMSO in the cultures is very small. In our opinion it does not influence the results obtained. This lack of effect is the cause of not including DMSO in the analysis of its radioprotective effect in the manuscript. However, we have determined the effect of high concentrations of DMSO on cell cultures, similar to those of the other substances tested, to determine it radioprotective capacity.

The following graphs show the results obtained with DMSO under the same experimental conditions and doses similar to the treatments administered. At these high concentrations, the radioprotective capacity of DMSO can be determined.

(Please see the attachment)

Comment:-Before administering any compounds to humans in an exposure scenario to ionizing radiation, studies should be performed on a wider range of cell lines. Especially since different types of tissues can act quite diversely to the same compound with/without radiation exposure.

-Applications in medical diagnostics: As described above, different types of risks should always be heightened against each other. Do not overgeneralize statements. The matter is, due to its nature, quite complex.

-When it is being written about certain concentration ranges (line 440) they should be given (with reference to the literature) and compared to the concentrations in this study.

-Point clearly out, that prior to any application during radiation therapy treatment further studies are needed. They are needed to make sure that the respective cancerous tissues do not benefit "more" from the protective effects than the healthy tissue(compare Rieckmann et al https://www.nature.com/articles/s41598-019-43040-w).

Response. We have added the following paragraph (line 470):

“Evidently, before administering any compounds to humans in an ionizing radiation exposure scenario to ionizing radiation, studies should be performed on a wider range of cell lines. Especially since different types of tissues can act quite diversely to the same compound with/without radiation exposure. Further, the effects of antioxidant supplements in oncology may be harmful. Although some studies have suggested that antioxidants can protect normal tissues from chemotherapy- or radiation-induced damage, others have claimed that supplementary antioxidants during chemotherapy and radiation therapy should be discouraged because they may actually protect the tumour cells and so reduce survival of the patient [8]”.

Comment:-The final statement of the discussion, that the compounds should be used even "against any dose of radiation however small they may be (line 457)" is to be seen very critical, and should be removed. The possible advertise effects of administration of all kind of drugs should be always weightened carefully against low-dose effects of radiation.

Response. The following paragraph has been deleted:

“These substances should be used against any dose of radiation however small they may be; and also they could be administered even after having occurred the exposure to ionizing radiation, as does it happen in the accidental exposures.”

-We have incorporated new references:

7. von Sonntag, C. Free-Radical-Induced DNA Damage and Its Repair. A chemical Perspective. Springer, Berlin: Germany,2016;1-45.

28. Hahn, M.B.; Meyer, S.; Schroter, M.; Kunte, H.; Solomun, T.; Sturn H. DNA protection by Ectoine from ionizing radiation: Molecular Mechanisms. Phys. Chem. Phys. Chem. 2017, 19, 25717-277122.

29. Rieckmann, T.; Gatzemeier, F.; Christiansen, S.; Rothkamm, K.; Münscher, A. The inflamation-reducing compatible solute ectoine does not impair the cytotoxic effect of ionizing radiation on head and neck cancer cells. Scientific Reports. 2019, 9, 6594-6601.

30. Buomino, E.; Schiraldi, C.; Baroni, A.; Paoletti, M.; De Rosa, M.;Tufano, M.A. Ectoine from halophilic microorganisms induces the expression of hsp70 and hsp70B′ in human keratinocytes modulating the proinflammatory response. Cell Stress & Chaperones, 2005, 10, 197-203.

33. Zhou, L.; Elias, R.J.Factors influencing the antioxidant and pro-oxidant    activity of polyphenols in oil-in-water emulsions. J Agric Food Chem 2012,  60, 2906-15.

34. Miura, T.; Muraoka, S.; Fujimoto, Y. Inactivation of creatine kinase induced by quercetin with horseradish peroxidase and hydrogen peroxide. pro-oxidative and anti-oxidative actions of quercetin. Food Chem Toxicol 2003,41, 759-65.

35. Raja, S.B.; Rajendiran, V.; Kasinathan, N.K.; P, A.; Venkatabalasubramanian, S.; Murali, M.R.; Devaraj, H.; Devaraj, S.N. Differential cytotoxic activity of Quercetin on colonic cancer cells depends on ROS generation through COX-2 expression. Food Chem Toxicol 2017,106, 92-106.

36. Dajas, F.; Abin-Carriquiry, J.A.; Arredondo, F.; Blasina, F.; Echeverry, C.; Martínez, M.; Rivera, F.; Vaamonde, L. Quercetin in brain diseases: Potential and limits. Neurochem Int 2015, 89,140-8.

Reviewer 2 Report

This manuscript describes a study that tests the effects of Pycnanthus angolensis Warb (African nutmeg) seed extract (PASE) on micronuclei formation, induced in human lymphocytes by ex vivo exposure to X-rays and mouse bone marrow after in vivo radiation. In addition, the effects of PASE on the performance of irradiated human and mouse cell lines in an MTT assay are assessed. PASE had radiation protective properties equally to or better than known radiation protectors and other flavonoids. Based on these results, the authors suggest that PASE may be a safe dietary substance to protect individuals from genotoxic effects of low dose radiation.

The development of safe radiation countermeasures is important, and the comparison of PASE with known potential countermeasures is helpful. However, relying only on the MTT assay to assess cell survival has its drawbacks. Also, some clarifications are required in the text and figures, as indicated below.

Specific comments

The results from MTT assays depend both on cell number and metabolic status of the cells. Therefore, a clonogenic cell survival assay is more commonly used to measure radiation-induced cell death and survival. In the current study, it would be good to select some of the treatment conditions that provide the main findings in the MTT assay and repeat the measurements in a clonogenic cell survival assay.

Erodictyol is usually spelled as eriodictyol.

Some clarifications are required in the methods section: Line 106: “Active compounds from different parts of the plants and seeds,” This phrase is not clear. It reads as if compounds were extracted from other parts of the Pycnanthus angolensis Warb plant (not just from the seeds), but there is no mention of testing other parts of the plant elsewhere in the manuscript. Line 132: “Two signals were acquired” but the text that follows this phrase seems to describe only one of the signals. Line 155: Please indicate the gender of each of the blood donors. Line 159: Please indicate at what time after radiation the micronucleus assay was performed. Line 168: Please indicate at what time before radiation the drinking water with test substances was administered to the mice. If immediately before irradiation, was there enough time for the substances to be absorbed and reach appropriate plasma levels during radiation? Line 172: Please indicate at what time after radiation the micronucleus assay was performed. Line 198: Please describe whether or not vehicle (DMSO?) was added to control wells. Line 204: Please indicate the filtration of the X-rays (copper/aluminum? and their thickness). Lines 208 and 210: Should FDO say FOD?

The statistical analysis section describes the calculation of a protection factor (PF) and dose reduction factor (DRF). However, these are briefly mentioned in the abstract but not in the results or discussion sections in the main text.

Figure 4: eriodictyol and quercetin seem to enhance radiation-induced micronucleus formation. Please discuss those results.

Figure 6: Please add a legend or describe the dashed and solid lines.

Figure 7: The lines in this figure are not clear. API cannot be seen in the 24 h panel. PASE cannot be seen in the 48 h panel. Also, should ROS in the legend say RA?

Author Response

Reviewer2.

This manuscript describes a study that tests the effects of Pycnanthus angolensis Warb (African nutmeg) seed extract (PASE) on micronuclei formation, induced in human lymphocytes by ex vivo exposure to X-rays and mouse bone marrow after in vivo radiation. In addition, the effects of PASE on the performance of irradiated human and mouse cell lines in an MTT assay are assessed. PASE had radiation protective properties equally to or better than known radiation protectors and other flavonoids. Based on these results, the authors suggest that PASE may be a safe dietary substance to protect individuals from genotoxic effects of low dose radiation.

The development of safe radiation countermeasures is important, and the comparison of PASE with known potential countermeasures is helpful. However, relying only on the MTT assay to assess cell survival has its drawbacks. Also, some clarifications are required in the text and figures, as indicated below.

Specific comments

The results from MTT assays depend both on cell number and metabolic status of the cells. Therefore, a clonogenic cell survival assay is more commonly used to measure radiation-induced cell death and survival. In the current study, it would be good to select some of the treatment conditions that provide the main findings in the MTT assay and repeat the measurements in a clonogenic cell survival assay.

Comment:-Erodictyol is usually spelled as eriodictyol.

Response. We have corrected this point.

Comment:-Some clarifications are required in the methods section: Line 106: “Active compounds from different parts of the plants and seeds,” This phrase is not clear. It reads as if compounds were extracted from other parts of the Pycnanthus angolensis Warb plant (not just from the seeds), but there is no mention of testing other parts of the plant elsewhere in the manuscript.

Response. We have corrected this point.

-The paragraph now reads (line 106): “Active compounds from different seeds of the plant (PASE) were extracted for”

Comment:- Line 132: “Two signals were acquired” but the text that follows this phrase seems to describe only one of the signals.

Response. The location of Figure 1 misled the interpretation of the results. It has now been properly placed in the text. We performed two different chromatographic analyzes that are described in the sections 2.4.1 and 2.4.2 of material and methods. respectively.

The first chromatographic analysis to perform an initial elucidation of active PASE compounds is described in Material & Methods 2.4.1. This first chromatogram corresponds to Figure 1, whose legend we have modified. The paragraph now reads: “Figure 1. Characteristic chromatogram (material and methods 2.4.1) of Pycnanthus angolensis Seed extract (PASE), monitored at 250 nm. (Peaks: (1) Rt 22.7 min, sargahydroquinoic acid; (2) Rt 24.8 min, sargaquinoic acid; (3) Rt 26.4 min, sargachromenol.)”

Subsequently, we performed a second chromatographic analysis to perform the determination and identification of assets by HPLC-MS, under the conditions described in section 2.4.2 of material and methods. These new chromatographic conditions (basic change of column and mobile phase) were necessary to obtain better results in this analytical determination. From this second chromatogram we have considered more clarifying to make a Table (Table 1) with the experimental results obtained from the PASE. 

We have modified the legend of Table1, now reads:” Table 1. Experimental results (material and methods 2.4.2) versus theoretical structural data of proposed compounds present in Pycnanthus angolensis Seed, monitored at 280 nm HPLC-MS.”

This chromatogram has allowed us to confirm the experimental results obtained by the first chromatographic analysis, but it is important to consider that the retention times (Rt) of the three isolated and identified compounds are different in both chromatographic profiles.

The chromatogram obtained is described in Table 1 is as follows:

(Please see the attachment:Table 1)

With this background we have modified Figure 1 and the text of Figure 1. Now it shows:

(Please see the attachment: Figure 1. )

Comment:-Line 155: Please indicate the gender of each of the blood donors.

Response.  We have corrected this point.

-The paragraph now reads (line 156): “Venous blood was obtain by venipuncture from the arm veins of three supposedly healthy young female donors into heparinized tubes.”

Comment:-Line 159: Please indicate at what time after radiation the micronucleus assay was performed.

Response. - We have corrected this point.

-The paragraph now reads (line 159):Immediately after irradiation with X-rays, the cytokinesis-block micronucleus (CBMN) assay as described by Fenech and Morley [13] and adapted by the International Atomic Energy Agency [14] was used to access damage in the cultured irradiated human lymphocytes.”

Comment:-Line 168: Please indicate at what time before radiation the drinking water with test substances was administered to the mice. If immediately before irradiation, was there enough time for the substances to be absorbed and reach appropriate plasma levels during radiation?

Response.  We have corrected this point.

-The paragraph now reads (line 169): “All solutions were prepared daily and the test substances were dissolved to a concentration of 0.2% in their drinking water during one week before to exposition to x rays.”

Comment:-Line 172: Please indicate at what time after radiation the micronucleus assay was performed.

Response.  We have corrected this point.

-The paragraph now reads (line 175): “The in vivo micronucleus assay was performed on the bone marrow of the mice, as described by Schmid [15]. Twenty-four hours after X-ray exposure, the numbers of micronucleated polychromatic erythrocytes (MNPCEs) among 1,000 PCEs per mouse were determined by three specialists in a double-blind study.”

Comment:- Line 198: Please describe whether or not vehicle (DMSO?) was added to control wells.

Response. Yes, all cell cultures are performed with a small concentration of DMSO. We have also studied the genoprotective capacities (figures 4 and 5) and radioprotective capacities of DMSO for two reasons: one, being an antioxidant substance with the characteristics of substance with sulfur; another, to be used in cell culture media usually. At high doses it has a slight genoprotective activity if it is administered before X-ray irradiation, but this genoprotection disappears if it is administered after exposure to X-rays, in a similar way to what we determine with the other sulfur substance (AMF) (lines 321 and 347).

In the radioprotective study, the final concentration of DMSO in the cultures is very small. In our opinion it does not influence the results obtained. This lack of effect is the cause of not including DMSO in the analysis of its radioprotective effect in the manuscript. However, we have determined the effect of high concentrations of DMSO on cell cultures, similar to those of the other substances tested, to determine it radioprotective capacity.

The following graphs show the results obtained with DMSO under the same experimental conditions and doses similar to the treatments administered. At these high concentrations, the radioprotective capacity of DMSO can be determined.

(Please see the attachment)

Comment: -Line 204: Please indicate the filtration of the X-rays (copper/aluminum? and their thickness). Lines 208 and 210: Should FDO say FOD?

Response: - We have corrected this point.

-The paragraph now reads (line 205): “An Andrex SMART 200E (Yxlon International, Hamburg, Germany) X-ray producing equipment with the following characteristics was used: 200 kV, 4.5 mA, filtration of 2.5 mm of Al and dose rate of 1.3 cGy/s at a focus-object distance (FOD) of 35 cm. The experiments were performed at room temperature. For the determination of in vitro genotoxicity, whole human blood samples were exposed to 2 Gy X-rays at FOD of 35 cm; while for the in vivo study, conscious and immobilized animals were whole body irradiated to a dose of 500 mGy at an FOD of 74 cm. For the determination of the radioprotective capacity, cell cultures grown in microplates were irradiated to different doses of X-rays (0, 4, 6, 8 and 10 Gy) at an FOD of 35 cm.”

Comment:-The statistical analysis section describes the calculation of a protection factor (PF) and dose reduction factor (DRF). However, these are briefly mentioned in the abstract but not in the results or discussion sections in the main text.

Response: Previously the sentence read (line 358) In the PNT2 cells, we established protection and dose reduction factors of 35.5% and

- This has been modified and the paragraph now reads (line 367):

In the PNT2 cells, we established a protection factor of 35.5% and a dose reduction factor of 2.5±0.12 respectively after 48 h of incubation and exposure to 10 Gy of radiation, whereas in the B16F10 melanoma cells a protection factor of 41.2% and DRF of 4±0.2 were observed for the same incubation period (48 h).”

In Discussion we have modified the paragraph. The paragraph now reads (line 456):

“We have not found references on the radioprotective activity of Pycnanthus angolensis. Our results on cell survival, protection factor (PF) and dose reduction factor (DRL) also confirm the radioprotective effect of PASE against cytotoxic damage induced by IR on normal prostate (traditionally considered as radiosensitive cells) and melanoma tumor (considered as radioresistant cells).”

Comment:-Figure 4: eriodictyol and quercetin seem to enhance radiation-induced micronucleus formation. Please discuss those results.

Response:

-The paragraph now reads (line 435): “We have previously shown that the reduction of micronuclei induced by ionizing radiation in biological systems cannot be directly ascribed to one single compound with a peculiar chemical structure although observed genoprotective effects induced by the presence of some compounds seem to be related to their antioxidant capacities and bioavailability in the cellular milieu [8]. Consequently, we observed that flavan-3-ols showed the greatest protective capacity of all polyphenols evaluated in our previous study [12], while other flavonoids known to have high antineoplastic and antiproliferative capacities showed lower antimutagenic capacities [31,32]. It is the situation of our results obtained in the treatments with Q and E since both present a flavonoid structure with a catechol group in the B ring that gives them a proxidant capacity, even at the low concentrations used in this study, and that when reacting with the radical superoxide can generate hydrogen peroxide, increasing its cytotoxic capacity [33-36].

In addition, this genoprotective capacity was also found to depend on the degree of polymerization and solubility of the substances assessed, since both modify their bioavailability [8,10,11]. We believe that all of the above equally relevant to PASE which we used as a genoprotective substance against IR-induced damage in this study”

Comment:-Figure 6: Please add a legend or describe the dashed and solid lines.

Response. - We have incorporated in the figure 6 the description of each curve. The legend of the figure has also been modified to reflecte the changes made. The modified figure is depicted as follows:

(Please see the attachment: figure 6)

Comment:-Figure 7: The lines in this figure are not clear. API cannot be seen in the 24 h panel. PASE cannot be seen in the 48 h panel. Also, should ROS in the legend say RA?

Response: - We have corrected this point. We have modified Figure 7 and its text.

(Please see the attachment: figure 7)

We have incorporated new references:

7. von Sonntag, C. Free-Radical-Induced DNA Damage and Its Repair. A chemical Perspective. Springer, Berlin: Germany,2016;1-45.

28. Hahn, M.B.; Meyer, S.; Schroter, M.; Kunte, H.; Solomun, T.; Sturn H. DNA protection by Ectoine from ionizing radiation: Molecular Mechanisms. Phys. Chem. Phys. Chem. 2017, 19, 25717-277122.

29. Rieckmann, T.; Gatzemeier, F.; Christiansen, S.; Rothkamm, K.; Münscher, A. The inflamation-reducing compatible solute ectoine does not impair the cytotoxic effect of ionizing radiation on head and neck cancer cells. Scientific Reports. 2019, 9, 6594-6601.

30. Buomino, E.; Schiraldi, C.; Baroni, A.; Paoletti, M.; De Rosa, M.;Tufano, M.A. Ectoine from halophilic microorganisms induces the expression of hsp70 and hsp70B′ in human keratinocytes modulating the proinflammatory response. Cell Stress & Chaperones, 2005, 10, 197-203.  

33. Zhou, L.; Elias, R.J.Factors influencing the antioxidant and pro-oxidant activity of polyphenols in oil-in-water emulsions. J Agric Food Chem 2012, 60, 2906-15.

34. Miura, T.; Muraoka, S.; Fujimoto, Y. Inactivation of creatine kinase induced by quercetin with horseradish peroxidase and hydrogen peroxide. pro-oxidative and anti-oxidative actions of quercetin. Food Chem Toxicol 2003,41, 759-65.

35. Raja, S.B.; Rajendiran, V.; Kasinathan, N.K.; P, A.; Venkatabalasubramanian, S.; Murali, M.R.; Devaraj, H.; Devaraj, S.N. Differential cytotoxic activity of Quercetin on colonic cancer cells depends on ROS generation through COX-2 expression. Food Chem Toxicol 2017,106, 92-106.

36. Dajas, F.; Abin-Carriquiry, J.A.; Arredondo, F.; Blasina, F.; Echeverry, C.; Martínez, M.; Rivera, F.; Vaamonde, L. Quercetin in brain diseases: Potential and limits. Neurochem Int 2015, 89,140-8.

Round 2

Reviewer 1 Report

The authors improved the manuscript substantially.

I found only the following minor point and issues:

Original comment:-MTT test: is there any information of the cellular uptake of the
different compounds? this is very important to access protective methods
(compare Rieckmann et al https://www.nature.com/articles/s41598-019-
43040-w)-
Response. We are not privy to this type of information regarding the
substances tested in relation to MTT.
Comment: This statement might be added to the manuscript.

Original comment:-it might be of interest to include a table or figure with the data
of reated/untreated cells even tough there was no significant difference
observed between treatments without irradiation.
Response. The concentrations of the tested substances that we have selected
for this study have no effect on the cell survival. All cultures treated and non-
irradiated were found within 100 ± 5% cell survival during the different
incubation periods and no significant differences induced by the treatments
have determined. Therefore, we had no interest in incorporation this information
into the manuscript.
Comment: I would recommand to add this information to the text.

Original comment: -dose reduction factors should be given with uncertainity
values
Response. We have incorporated these values. Now the paragraph reads. (line
367): “In the PNT2 cells, we established a protection factor of 35.5% and a dose
reduction factor of 2.5±0.12 respectively after 48 h of incubation and exposure to 10 Gy
of radiation, whereas in the B16F10 melanoma cells a protection factor of 41.2% and
DRF of 4±0.2 were observed for the same incubation period (48 h)”.
Comment: Values and uncertainities throughout the manuscript should be all given with the same accuracy (e.g. 4.0+-0.3 not 4+-0.3)

Original comment:Discussion:
This part is the most critical one and needs careful revision.
- line 372: formulation is a little unclear: ROS, especially the most
important one, OH radicals, are proudced after ionization of water and
subsequent reactions (compare von sonntag).
Response. We have modified that aspect of the paragraph, which now reads
(line 385):
“X-ray exposure produces a massive generation of reactive oxygen species/free
radicals (ROS) in vivo. These ROS are formed by a sequential mechanism of electron
transfer, through which molecular oxygen successively gives rise to a superoxide
radical, hydrogen peroxide and hydroxyl radical [12,20-22]. In general, ionizing
radiation produces in DNA and its model systems a large number of different radicals
• •through the action of OH, eaq and H [7]. Its high reactivity produces an immediate
reaction in the vicinity of its generation; However, when its generation is massive as a
result of exposure to X-rays, the cytotoxic effect is no longer only local but can spread
through reactive species and other radicals within the intracellular and even
extracellular environment, increasing interaction with cellular phospholipids structures
of and inducing lipid peroxidation processes that increase the oxidative damage of DNA
[20,21]
Comment: Just a remark regarding the formulations above: OH radicals are mostly produced by water radiolysis. No molecular oxygen is needed in this reaction.
Please make the disctinction clear.

Original comment: -line 411: that radiation of same LET causes similar amount of
biological damage is not really surprising (compare work of Kellerer
"Fundamentals of Microdosimetry" 1985). since this study only includes
one x ray energy to state the this study confirm something about the LET
concept is a little bold. I would recommand reformulating this phrase
slightly.
...
All this would explain, therefore, that the genoprotective capacity of some substances
tested in this study be similar to either of both ionizing agents (X-rays and radiation
gamma [8]).”
Comment: To make it more readable I would reformulate the phrase as follows:
"This would explain that the genoprotective capacity of some substances
tested in this study are similar to xrays as well as
gamma radiation[8].”

Original comment:-How does the interpretation as "protection by OH scavenging"
can be explained when DMSO is much worse, which is known as a very
effective OH scavenger? Please discuss this point.
Response. We have also studied the genoprotective capacities (figures 4 and
5) and radioprotective capacities of DMSO for two reasons: one, being an
antioxidant substance with the characteristics of substance with sulfur; another,
to be used in cell culture media usually. At high doses it has a slight
genoprotective activity if it is administered before X-ray irradiation, but this
genoprotection disappears if it is administered after exposure to X-rays, in a
similar way to what we determine with the other sulfur substance (AMF) (lines
321 and 347).
In the radioprotective study, the final concentration of DMSO in the cultures is
very small. In our opinion it does not influence the results obtained. This lack of
effect is the cause of not including DMSO in the analysis of its radioprotective
effect in the manuscript. However, we have determined the effect of high
concentrations of DMSO on cell cultures, similar to those of the other
substances tested, to determine it radioprotective capacity.
The following graphs show the results obtained with DMSO under the same
experimental conditions and doses similar to the treatments administered. At
these high concentrations, the radioprotective capacity of DMSO can be
determined.
Comment: That the protective effect of DMSO is only visible when administered before radiation is in agreement with the proposed mechanism as OH scavenger, what is indeed known for DMSO. When the compounds studied are more effective than DMSO in terms of radioprotection, then it is most likely that other effects besides OH scavenging are responsible for the beneficial outcome.

Author Response

Reviewer 1

The authors improved the manuscript substantially.

I found only the following minor point and issues:

Original comment:-MTT test: is there any information of the cellular uptake of the different compounds? this is very important to access protective methods (compare Rieckmann et al https://www.nature.com/articles/s41598-019-43040-w)-

Response. We are not privy to this type of information regarding the substances tested in relation to MTT.

Comment: This statement might be added to the manuscript.

Response: We have added the following paragraph to the manuscript (line 501):

“We have not found specific information on the cellular uptake of the different compounds in relation to the micronucleus or MTT assays. We have previously shown that the reduction of micronuclei induced by ionizing radiation in biological systems cannot be directly ascribed to one single compound with a peculiar chemical structure although observed genoprotective effects induced by the presence of some compounds seem to be related to their antioxidant capacities and bioavailability in the cellular milieu [8].

Original comment:-it might be of interest to include a table or figure with the data of reated/untreated cells even tough there was no significant difference observed between treatments without irradiation.

Response. The concentrations of the tested substances that we have selected for this study have no effect on the cell survival. All cultures treated and non-irradiated were found within 100 ± 5% cell survival during the different incubation periods and no significant differences induced by the treatments have determined. Therefore, we had no interest in incorporation this information into the manuscript.

Comment: I would recommand to add this information to the text.

Response: Previously this paragraph read (line 413): “No significant differences were determined in the percentage cell survivals of the un-treated-cells and the cells treated with the different substances, demonstrating absence of toxicity effects of the substances administered.”

We have included the following paragraph into the text: “The concentrations of the tested substances that we selected for this study have no effect on the cell survivals. All cultures which were treated with the test substances but not exposed to irradiated were found to have cell survivals of within 100 ± 5% for the different incubation periods. Moreover, no significant differences in cell survival induced by the treatments was established, demonstrating absence of toxic effects of the substances administered.”

Original comment: -dose reduction factors should be given with uncertainity values

Response. We have incorporated these values. Now the paragraph reads. (line 367): “In the PNT2 cells, we established a protection factor of 35.5% and a dose reduction factor of 2.5±0.12 respectively after 48 h of incubation and exposure to 10 Gy of radiation, whereas in the B16F10 melanoma cells a protection factor of 41.2% and DRF of 4±0.2 were observed for the same incubation period (48 h)”.

Comment: Values and uncertainities throughout the manuscript should be all given with the same accuracy (e.g. 4.0+-0.3 not 4+-0.3)

Response: We have modified these values and the paragraph now reads (line 424): “In the PNT2 cells, we established a protection factor of 35.5% and a dose reduction factor of 2.5±0.2 respectively after 48 h of incubation and exposure to 10 Gy of radiation, whereas in the B16F10 melanoma cells a protection factor of 41.2% and DRF of 4±0.2 were observed for the same incubation period (48 h).”

Original comment:Discussion: This part is the most critical one and needs careful revision.

- line 372: formulation is a little unclear: ROS, especially the most important one, OH radicals, are proudced after ionization of water and subsequent reactions (compare von sonntag).

Response. We have modified that aspect of the paragraph, which now reads

(line 385): “X-ray exposure produces a massive generation of reactive oxygen species/free radicals (ROS) in vivo. These ROS are formed by a sequential mechanism of electron transfer, through which molecular oxygen successively gives rise to a superoxide radical, hydrogen peroxide and hydroxyl radical [12,20-22]. In general, ionizing radiation produces in DNA and its model systems a large number of different radicals

•through the action of OH, eaq and H [7]. Its high reactivity produces an immediate reaction in the vicinity of its generation; However, when its generation is massive as a result of exposure to X-rays, the cytotoxic effect is no longer only local but can spread through reactive species and other radicals within the intracellular and even extracellular environment, increasing interaction with cellular phospholipids structures of and inducing lipid peroxidation processes that increase the oxidative damage of DNA [20,21]

Comment: Just a remark regarding the formulations above: OH radicals are mostly produced by water radiolysis. No molecular oxygen is needed in this reaction.

Please make the disctinction clear.

Response: We have tried to clarify this situation. Now the paragraph reads (line 450): “In general, ionizing radiation produces in the vicinity of DNA and its environs a large number of different radicals such as •OH, e-aq and H• [7] which are mostly produced by the radiolysis water even in the absence of molecular oxygen.

Original comment: -line 411: that radiation of same LET causes similar amount of biological damage is not really surprising (compare work of Kellerer "Fundamentals of Microdosimetry" 1985). since this study only includes one x ray energy to state the this study confirm something about the LET concept is a little bold. I would recommand reformulating this phrase slightly. All this would explain, therefore, that the genoprotective capacity of some substances tested in this study be similar to either of both ionizing agents (X-rays and radiation gamma [8]).”

Comment: To make it more readable I would reformulate the phrase as follows:

"This would explain that the genoprotective capacity of some substances tested in this study are similar to x rays as well as gamma radiation [8].”

Response: We have modified this phrase. Previously it read as (line 499): “All this would explain, therefore, that the genoprotective capacity of some substances tested in this study be similar to either of both ionizing agents (X-rays and radiation gamma [8].”

Now the paragraph reads: “This would explain that the genoprotective capacity of some substances tested in this study are similar to x rays as well as gamma radiation [8].”

Original comment:-How does the interpretation as "protection by OH scavenging" can be explained when DMSO is much worse, which is known as a very effective OH scavenger? Please discuss this point. Response. We have also studied the genoprotective capacities (figures 4 and 5) and radioprotective capacities of DMSO for two reasons: one, being an antioxidant substance with the characteristics of substance with sulfur; another, to be used in cell culture media usually. At high doses it has a slight genoprotective activity if it is administered before X-ray irradiation, but this genoprotection disappears if it is administered after exposure to X-rays, in a similar way to what we determine with the other sulfur substance (AMF) (lines 321 and 347).

In the radioprotective study, the final concentration of DMSO in the cultures is very small. In our opinion it does not influence the results obtained. This lack of effect is the cause of not including DMSO in the analysis of its radioprotective effect in the manuscript. However, we have determined the effect of high concentrations of DMSO on cell cultures, similar to those of the other substances tested, to determine it radioprotective capacity.

The following graphs show the results obtained with DMSO under the same experimental conditions and doses similar to the treatments administered. At these high concentrations, the radioprotective capacity of DMSO can be determined.

Comment: That the protective effect of DMSO is only visible when administered before radiation is in agreement with the proposed mechanism as OH scavenger, what is indeed known for DMSO. When the compounds studied are more effective than DMSO in terms of radioprotection, then it is most likely that other effects besides OH scavenging are responsible for the beneficial outcome.

Response: Effectively, there must be other variables that contribute to improve this radioprotective capacity, but for now we have not been able to make further progress in this regard. Thank you very much for your help and effort to improve our manuscript.

Reviewer 2 Report

The authors have responded to many of my prior critiques. A few comments remain:

The results from MTT assays depend both on cell number and metabolic status of the cells. Therefore, a clonogenic cell survival assay is more commonly used to measure radiation-induced cell death and survival. In the current study, it would be good to select some of the treatment conditions that provide the main findings in the MTT assay and repeat the measurements in a clonogenic cell survival assay.
In the methods section, please describe whether DMSO was added to control cell cultures. From the authors’ response to my prior critiques, I do not understand what was done as part of this study and what was done in the past. According to the manuscript, lines 321-322, DMSO was indeed added to cell cultures and separately tested for radiation protection. But nothing is mentioned in the methods. If DMSO was NOT added, then please reword lines 321-322.
Line 169: The English grammar makes it difficult to understand this sentence. Please reword into: “starting one week before X-ray exposure.”
Figure 7: The top panel has a light grey line with circular symbol (a horizontal line at 100% cell survival) that is not described in the legend.
Line 402: Please clarify “eaq”
Line 460: Please add a sentence to explain why this text was added. Something like: “This may explain the increase in micronuclei formation after treatment with Q and E.”

Author Response

Reviewer 2

The authors have responded to many of my prior critiques. A few comments remain:

The results from MTT assays depend both on cell number and metabolic status of the cells. Therefore, a clonogenic cell survival assay is more commonly used to measure radiation-induced cell death and survival. In the current study, it would be good to select some of the treatment conditions that provide the main findings in the MTT assay and repeat the measurements in a clonogenic cell survival assay.

Response: As our reviewer pointed out, we controlled the initial number of cells in each microplate during the incubation periods to assess the metabolic states or activities of the cells in each assay. Following these signals, we intend to confirm these results using a clonogenic cell survival assay in a subsequent study.

In the methods section, please describe whether DMSO was added to control cell cultures. From the authors’ response to my prior critiques, I do not understand what was done as part of this study and what was done in the past. According to the manuscript, lines 321-322, DMSO was indeed added to cell cultures and separately tested for radiation protection. But nothing is mentioned in the methods. If DMSO was NOT added, then please reword lines 321-322.

Now I understand your comment. In the study of genoprotection, we tested DMSO as an antioxidant. DMSO was added to blood samples in the micronucleus assay in the irradiated human lymphocytes blocked with Cyt B ("in vitro" Micronucleus assay). In this test we demonstrated that DMSO shows a genoprotective effect similar to AMF (as do sulfur-containing compounds). Therefore, it is described in line 103.

In the radioprotection study (MTT assay), DMSO was not added to PNT2 or to B16F10 cell cultures. In this manuscript we have not shown the effect of DMSO on these cells in comparison with the effects of PASE because we have presented other substances of greater interest. However, although we have not presented the radioprotective effect of DMSO on PNT2 and B16F10 cell cultures in this manuscript, we determined this effect. The DMSO graphs shown in the previous response correspond to our unpublished results of the radioprotective effect of high concentrations of DMSO on PNT2 and B16F10 cells and which also confirm their radioprotective capacity.

The description in the noted paragraph (line 321-322) is correct, although a bit ambiguous. We have corrected it as following:

--(line 335-337) The chemical structures of the main flavonoids and polyphenols in the different extracts used in this study and the chemical structures of sulfur-containing compounds (Amifostin and DMSO) are shown in Figure 3.

-- Before the paragraph read:” Figure 3. Chemical structures of flavonoids and polyphenols present in the tested extracts.”

Now the paragraph says: Figure 3. Chemical structures of different substances tested in this study.

-Before the paragraph read (line 396-402: “However, when the different substances were administered after X-ray exposure, the MN frequencies were higher than what was observed in the pre X-ray treatments. It is clear that while CA showed significant antimutagenic activity, RA demonstrated a low degree of radioprotective activity, and DMSO along with AMF (sulfur-containing compounds) lost their radioprotective capacities against X-ray

Now the paragraph says: “However, when the different substances were administered after X-ray exposure, the MN frequencies were higher than what was observed in the pre X-ray treatments. It is clear that while CA showed significant antimutagenic activity, RA demonstrated a low degree of genoprotective activity, and DMSO along with AMF (sulfur-containing compounds) lost their genoprotective capacities against X-ray.”

Line 169: The English grammar makes it difficult to understand this sentence. Please reword into: “starting one week before X-ray exposure.”

Previously the sentence read: “All solutions were prepared daily and the test substances were dissolved to a concentration of 0.2% in their drinking water before one week before to exposition to x ray 

- The paragraph now reads (line 174): “All solutions were prepared daily and the test substances were dissolved to a concentration of 0.2% in their drinking water. This treatment commenced one week prior to x-ray exposure.”

Figure 7: The top panel has a light grey line with circular symbol (a horizontal line at 100% cell survival) that is not described in the legend.

Response: The indicated curve corresponds to the Apigenin (API). We have corrected graph 7. Now graph 7 shows:

(Please see the attachment)

Line 402: Please clarify “eaq

Response: We have modified the symbol. Now read: “e-aq

Discovered and studied by radiation chemists, the hydrated electron, a unique ion, provides great potential for interpreting reaction mechanisms in chemistry and biology.  Hidrated electrons, designated by the symbol e-aq , result from secondary electrons generated in water by ionizing radiations. Existence of a hydrated electron as a byproduct of water radiolysis was established more than 50 years ago, yet this species continues to attract significant attention due to its role in radiation chemistry, including DNA damage, and because questions persist regarding its detailed structure. In water, or even any polar liquid, the secondary electron cloud, is trapped by the solvent molecules to form another class of electronic structures, the solvated electrons, sometimes called aquous electrons (eaq-). These trapped electrons have mobility inside the liquid medium determined by the physicochemical nature of the liquid. The radiolysis of water can be summarized in one equation:

H2O +  radiation----à e-aq - + OH• + H• + H2 + H2O2

Line 440: Please add a sentence to explain why this text was added. Something like: “This may explain the increase in micronuclei formation after treatment with Q and E.” 

The paragraph now reads (line 505-509): “This may explain the increase in micronuclei formation obtained after treatments with Q and E since both present a flavonoid structure with a catechol group in the B ring that gives them proxidant capacities, even at the low concentrations used in this study. When reacted with the superoxide radical can lead to the generation of hydrogen peroxide, thus increasing its cytotoxic capacity [33-36]”

This manuscript is a resubmission of an earlier submission. The following is a list of the peer review reports and author responses from that submission.